# Unraveling metal effects on $CO_2$ uptake in pyrene-based metal-organic frameworks

Nency P. Domingues [1], Miriam J. Pougin[1], Yutao Li[1], Elias Moubarak[1], Xin Jin [1], F. Pelin Uran[1], Andres Ortega-Guerrero[1,2], Christopher P. Ireland[1], Pascal Schouwink[3], Christian Schürmann [4], Jordi Espín [5], Emad Oveisi [6], Fatmah Mish Ebrahim[1,7], Wendy Lee Queen[5] & Berend Smit [1] ✉

Pyrene-based metal-organic frameworks (MOFs) have tremendous potential for various applications. With infinite structural possibilities, the MOF community often relies on simulations to identify the most promising candidates for given applications. Among thousands of reported structures, many exhibit limited reproducibility − in either synthesis, performance, or both − owing to the sensitivity of synthetic conditions. Geometric distortions that may arise in the functional groups of pyrene-based ligands during synthesis and/or activation cannot easily be predicted. This sometimes leads to discrepancies between in silico and experimental results. Here, we investigate a series of pyrene-based MOFs for carbon capture. These structures share the same ligand (1,3,6,8−tetrakis(p−benzoic acid)pyrene (TBAPy)) but have different metals (M-TBAPy, M = Al, Ga, In, and Sc). The ligands stack parallel in their orthorhombic crystal structure, creating a promising binding site for $CO_2$. As predicted, the metal is shown to affect the pyrene stacking distance and, therefore, the $CO_2$ uptake. Here, we investigate the metal's intrinsic effects on the MOFs' crystal structure. Crystallographic analysis shows the emergence of additional phases, which thus impacts the overall adsorption characteristics of the MOFs. Considering these additional phases improves the prediction of adsorption isotherms, enhancing our understanding of pyrene-based MOFs for carbon capture.

Metal-organic frameworks (MOFs) can be synthesized with various metal ions and organic ligands with different or similar topologies and chemistries[1]. Because of their easy tunability and chemical stability, they have become relevant for a variety of applications, including gas storage and separation, catalysis, drug delivery, sensing, and gas adsorption[2–6].

While countless MOFs have been reported as promising for carbon capture because of their selective $CO_2$:$N_2$ uptake and chemical stability[5,7–11], their higher affinity for $H_2O$ compared to $CO_2$ has often been a bottleneck for wet flue gas carbon capture. Boyd et al.[12] demonstrated that MOFs with aromatic ligands arranged in parallel stacks are promising for $CO_2$ binding. In contrast, the $H_2O$ adsorption

[1]Laboratory of Molecular Simulation (LSMO), Institut des Sciences et Ingénierie Chimiques, École Polytechnique Fédérale de Lausanne (EPFL), Rue de l'Industrie 17, 1951 Sion, Switzerland. [2]Nanotech@surfaces Laboratory, Empa - Swiss Federal Laboratories for Materials Science and Technology, 8600 Dübendorf, Switzerland. [3]X-ray Diffraction and Surface Analytics Platform, École Polytechnique Fédérale de Lausanne (EPFL), Rue de l'Industrie 17, 1951 Sion, Switzerland. [4]Rigaku Europe SE, Hugenottenallee 167, 63263 Neu-Isenburg, Germany. [5]Laboratory for Functional Inorganic Materials (LFIM), Institut des Sciences et Ingénierie Chimiques, École Polytechnique Fédérale de Lausanne (EPFL), Rue de l'Industrie 17, 1951 Sion, Switzerland. [6]Interdisciplinary Centre for Electron Microscopy (CIME), École Polytechnique Fédérale de Lausanne (EPFL), 1015 Lausanne, Switzerland. [7]Cavendish Laboratory, School of Physical Sciences, University of Cambridge, Cambridge, United Kingdom. ✉e-mail: berend.smit@epfl.ch

exhibits relatively low Henry coefficients, which implies a lower affinity for the adsorbate. The parallel aromatic rings offer a nearly ideal interaction with all three atoms of the $CO_2$ molecule, while the binding energy of $H_2O$ is restricted by the absence of hydrogen-bonding sites[12]. This preference for $CO_2$ over $H_2O$ makes such MOFs suitable for wet flue gas carbon capture. The authors propose that the optimal spacing between ligand's stacks for $CO_2$ capture follows a volcano trend: if the spacing is too small, $CO_2$ does not fit between the stacks − almost no (or negligible) uptake− whereas if it is too large, the $\pi$-$CO_2$ interactions would be too weak, and would not retain much $CO_2$. However, an intermediate distance would provide the optimal uptake. Their study suggested that the ideal range for the stacking distance would be between 6.5 - 7.0 Å. This could be obtained if aluminum or gallium were used as the metal centers, leading to higher uptakes. On the other hand, indium or scandium would lead to a larger distance and thus lower uptake[12].

In this work, we conduct a combined experimental and computational study of a series of pyrene-based MOFs for carbon capture. All structures possess the same ligand (i.e., 1,3,6,8−tetrakis(p−benzoic acid)pyrene (TBAPy)) (Fig. 1), but different metals (i.e., M = aluminum (Al), gallium (Ga), scandium (Sc), and indium (In)), and we refer to them as "M-TBAPy". The selection of TBAPy as the ligand is based on the promising nature of the parallel aromatic rings of pyrene for $CO_2$ capture[9,10,12,13]. The impact of metal substitution has been previously investigated for different families of MOFs in the context of $CO_2$ capture[14–17], vapor adsorption[17], as well as gas separation[18–20]. In this study, we analyze a particular class, pyrene-based MOFs, in dry and wet conditions, and focus on the influence of these substitutions on the $CO_2$ uptakes and adsorption properties of these particular MOFs, where the distance between pyrene stacks is tuned depending on the metal incorporated in the structure. Moreover, given the possible different phases present in some of the MOFs studied (i.e., monoclinic and orthorhombic), we also demonstrate the effect that each of them has on the adsorption behavior of the materials, which challenges the predictions made by Boyd et al.[12]. While their computational study provides valuable insights for future design of sorbent materials, here we aim to highlight the importance of experimental validation, as small variations in the crystalline phase can lead to considerably different adsorption characteristics.

**Fig. 1 | Ligand's chemical structure.** Chemical structure of 1,3,6,8−tetrakis-(p−benzoic acid)pyrene (TBAPy), with the pyrene core highlighted in red.

Moreover, we experimentally and computationally investigate the performance of a mixed-cation TBAPy-based MOF. We explore different Al vs Sc metal arrangements within the same structure, which induce nonparallel pyrene stacks, and thus affect the critical adsorption sites of $CO_2$. Here, we aim to elucidate how these local structural modifications influence the uptake of $CO_2$, $N_2$, and $H_2O$, and thus provide insights into the adsorption behavior of these materials.

## Results
### Materials syntheses
In this work, we synthesize four different pyrene-based MOFs, with general formula $M_2(OH)_2$(TBAPy) (with M = Al(III), Ga(III), Sc(III), and In(III)). The synthesis of the ligand is based on two reported procedures[21,22] (Supplementary Note 1). The syntheses of Al-TBAPy[12], Sc-TBAPy[23], In-TBAPy[22], and Ga-TBAPy[24,25] are also reported.

These MOFs can be synthesized by mixing the metal precursor (i.e., M(NO$_3$)$_3$ · xH$_2$O, with M = Al, Sc, In or Ga, 0.03 mmol) with the ligand (i.e., TBAPy, 0.02 mmol) in a solvent mixture of dimethylformamide (DMF)/dioxane/$H_2O$ (4 mL, ratio 2/1/1), and concentrated hydrochloric acid (HCl) (32 %, 10 μL) as modulator. Once all precursors are fully dissolved, the 12 mL Pyrex glass vials are heated to 85 °C for 12 h, yielding crystalline products. All MOFs are extensively characterized using powder X-ray diffraction (PXRD), $N_2$ isotherms at 77 K, thermogravimetric analysis (TGA), and $CO_2$, $N_2$ and $H_2O$ isotherms at 40 °C. Throughout this work, the experimental data is compared to simulation results based on DFT-optimized models of different crystallographic phases of the M-TBAPy MOFs. Details on the computational methods and the various structural models are discussed in Supplementary Note 2.

### Structure description and powder x-ray diffraction analysis
We begin the presentation of our results with Al- and Sc-TBAPy, as these maintain the orthorhombic *Cmmm*-phase after synthesis and the activation process required for $CO_2$, $N_2$ and $H_2O$ adsorption isotherms. In Al- and Sc-TBAPy, the metal ions form a metal-oxide backbone of 1D chains of octahedral $MO_4(OH)_2$ units (with M = Al(III) and Sc(III)), wherein each M(III) metal ion is coordinated to four TBAPy ligands and two $\mu_2$ trans-hydroxide anions (Fig. 2a–c). This coordination environment matches the orthorhombic structure considered in the computational simulations by Boyd et al.[12], where the pyrene ligands stack parallel along the b-axis (Fig. 2b). This arrangement creates the highly favorable adsorption site for $CO_2$ in the presence of $H_2O$, as highlighted in their study[12] (site A, in Fig. 2b). This framework also features infinite rhombic channels (i.e., B and C in Fig. 2c) along the b-crystallographic axis. The PXRD patterns of the as-made and activated Al- and Sc-TBAPy MOFs are in good agreement with the predicted PXRD profiles of the simulated orthorhombic structures (Fig. 3a, b), making them isostructural MOFs. Synchrotron ex-situ PXRD measurements on the activated versions of both MOFs provide sufficient quality to perform Rietveld analysis. These experiments show that an orthorhombic *Cmmm*-phase is present as the primary phase in these MOFs (Supplementary Note 3).

The as-made In-TBAPy follows the same coordination motif as described for Al- and Sc-TBAPy, and its PXRD fully matches the simulated orthorhombic structure directly collected from Stylianou et al.[22] crystallographic data (Fig. 3c). Stylianou et al.[22] report a structural change and loss of crystallinity upon solvent removal. Similarly, in this study, we observe the same monoclinic distortion, evidenced in the PXRD with the emergence of additional broad peaks at around $2\theta \approx 9.2°$ and 9.9° upon guest loss. Synchrotron ex-situ PXRD measurement performed on activated In-TBAPy shows that an orthorhombic *Cmmm*-phase is present as the primary phase (Supplementary Note 3). In the as-made o-In-TBAPy, the pyrene ligand stacks perfectly along the b-axis, similarly to the orthorhombic structures of Al- and Sc-TBAPy (Fig. 2b). The activated material's PXRD profile is indexed with a

Orthorhombic Structure                 Monoclinic Structure

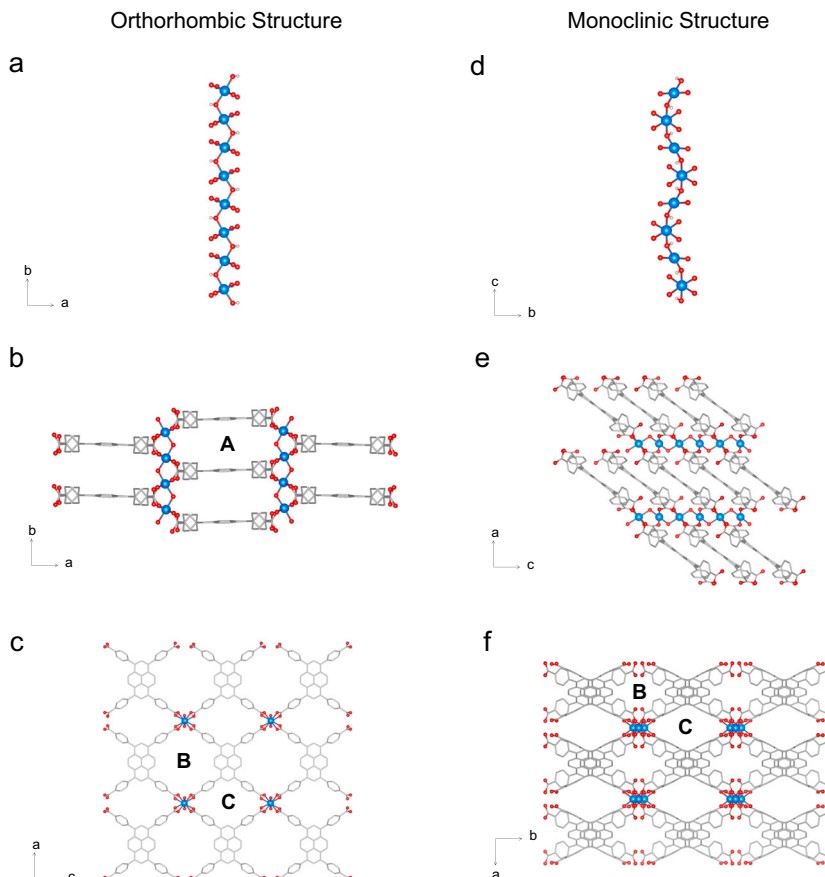

**Fig. 2 | Characteristic orthorhombic and monoclinic structures.** Crystal structure representation of the M-TBAPy MOFs for each system: orthorhombic (characteristic of as-made and activated Al- and Sc-TBAPy, as well as the as-made In-TBAPy) and monoclinic (characteristic of as-made and activated Ga-TBAPy, and activated In-TBAPy). Inorganic backbone characteristic of (**a**) the orthorhombic and

(**d**) monoclinic unit cells, highlighting the buckled nature of the latter. Different views for the (**b**, **c**) orthorhombic and (**e**, **f**) monoclinic unit cells. These figures highlight the pyrene stacks along the *b*-axis and the three types of sites (A, B, and C) present in the structures. Color code: C (gray), O (red), metal (light blue). Carbons (C) and hydrogens (H) are hidden for clarity apart from sub-figures (**a**) and (**d**).

primitive monoclinic cell (Fig. 3c). The simulated monoclinic model of In-TBAPy (which we will refer to as *m*-In-TBAPy), no longer shows this perfect stacking but rather a distorted structure with a buckled metal rod (Fig. 2d–f).

In this work, we also discuss Ga-TBAPy[24,25]. Despite numerous efforts, the synthesis of Ga-TBAPy produces two different polymorphs. Alongside the orthorhombic *Cmmm*-phase, *o*-Ga-TBAPy, we observe the appearance of broad Bragg peaks in the as-made sample (Supplementary Note 4) at $2\theta \approx 7.6°$ and 9.5°, indicating the presence of the monoclinic phase in the as-synthesized MOF. It is noted that by increasing the time and temperature of the synthesis, the peaks associated with the monoclinic phase are enhanced. To comprehensively characterize this phase, the MOF is synthesized at 120 °C for 48h (Fig. 3d). This structure is first indexed ab initio, using synchrotron powder diffraction data, with a primitive monoclinic cell with a monoclinic angle close to 90°. Due to the sample's complexity and poor data quality, the structure solution in real space was unsuccessful. The structure was then solved using micro-electron diffraction (microED/3D-ED). MicroED/3D-ED confirms the determined monoclinic cell and allows to assign the space group symmetry *P*2/*c* with unit cell parameters $a = 11.1(4)$ Å, $b = 15.1(3)$ Å and $c = 12.3(4)$ Å, $\beta = 90.12(7)°$ (Supplementary Note 5). This polymorph is referred to as *m*-Ga-TBAPy. The intensity of the characteristic monoclinic peaks increases upon activation, suggesting, in turn, a structural distortion with respect to the parent *Cmmm*-phase. To account for the different phases that we observe experimentally in In- and Ga-TBAPy, two computational models are developed (Supplementary Note 2): when we superimpose

the simulated patterns of both models, they collectively reproduce the experimental peaks in the activated materials. This observation indicates the coexistence of two distinct phases within the activated experimental structure used for $CO_2$, $N_2$ and $H_2O$ isotherms at 40 °C. Confirming our hypothesis of two phases present in the activated Ga-TBAPy, Rietveld refinement of the synchrotron ex-situ PXRD measurements reveals that, although an orthorhombic *Cmmm*-phase is present, a monoclinic *mcl*-phase coexists (Supplementary Note 3). However, given the complexity and quality of the data, additional phases may not be excluded.

Symmetry-lowering from the parent phase leads to two independent Ga atoms in *m*-Ga-TBAPy, that form a buckled chain of edge-sharing octahedra along the crystallographic *c* axis (Fig. 2d–f). This implies bond reconstruction from the corner-sharing chains in *o*-Ga-TBAPy. For the orthorhombic structure, we observe an inter-aromatic spacing of 6.65 Å, while for the *m*-phase 3.77 Å (Table 1), due to a pronounced distortion of the general topology. This distortion might be the cause of the peak broadening in the powder diffraction pattern. Figure 3d shows that, qualitatively, both as-made and activated materials are in good agreement with the monoclinic cell solved by microED/3D-ED (i.e., "Simulated Monoclinic"). This is most easily seen by the enhancement of the peaks at $2\theta \approx 7.6°$ and 9.5°, similar to In-TBAPy. Rietveld refinements are performed on Ga-data but can serve merely as validation since data quality does not allow us to refine the actual structure. The Bragg peaks of *m*-Ga-TBAPy exhibit anisotropic characteristics. Among these peaks, h00 reflections demonstrate the highest level of broadening, while 0k0 reflections exhibit the lowest

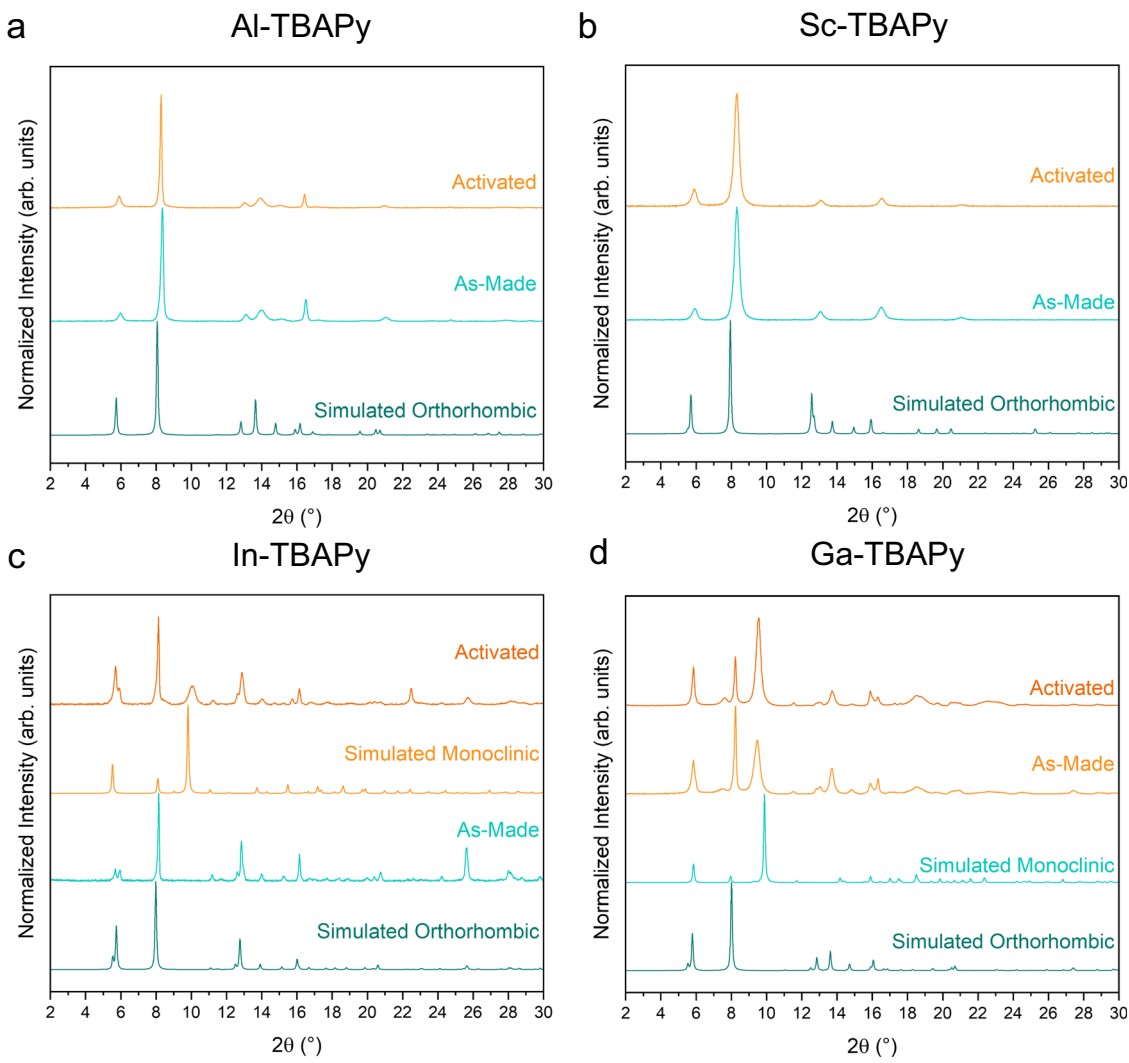

**Fig. 3 | Powder x-ray diffraction (PXRD) patterns.** Experimental as-made and activated PXRD patterns of all M-TBAPy MOFs, with the simulated patterns derived from the respective structure models. (**a**) Al-TBAPy, (**b**) Sc-TBAPy, (**c**) In-TBAPy, and (**d**) Ga-TBAPy ($\lambda$ = 1.5406 Å). Source data are provided as a Source Data file.

## Table 1 | Key characteristics of the MOFs

| MOF | Crystal system | Predicted inter-aromatic spacing (Å) | M–O$_{Ligand}$ distance (Å) | M–O–M Angle (°) |
| --- | --- | --- | --- | --- |
| Al-TBAPy | Orthorhombic | 6.63 | 1.94 | 125 |
| Ga-TBAPy | Orthorhombic | 6.65 | 2.04 | 118 |
| Ga-TBAPy | Monoclinic | 3.77 | 2.09 | 108 |
| In-TBAPy | Orthorhombic | 7.12 | 2.20 | 111 |
| In-TBAPy | Monoclinic | 4.00 | 2.23 | 101 |
| Sc-TBAPy | Orthorhombic | 7.23 | 2.14 | 120 |

Predicted inter-aromatic spacing, M–O$_{Ligand}$ distances and M–O–M angles (with M = Al, Ga, In, and Sc) derived from our optimized CIFs.

broadening. This would agree with the structural distortion, which affects the structure the least along the *b*-axis.

To further monitor the evolution of this distortion, synchrotron in-situ variable temperature PXRD under dynamic vacuum is also performed on the Ga-system. We describe a potential sequence of events based on the data obtained (Supplementary Note 6). A structural rearrangement at 91 °C is followed by cell expansion at 423 °C, which coincides with the vanishing of several diffraction peaks (Supplementary Note 6). The structural re-arrangement and possible distortion of the lattice are irreversible events. Upon cooling the sample to room temperature, the MOF can be indexed with orthorhombic cell parameters, similar to the as-made materials of Al- and Sc-TBAPy.

The monoclinic structures of Ga-TBAPy and In-TBAPy, no longer show the perfect stacking of the TBAPy ligand as observed in the orthorhombic Al- and Sc- structures (Fig. 2d–f). In the orthorhombic *Cmmm*-structure, the carboxylate groups of the pyrene ligand are fully deprotonated and bound to the metal center. Each metal center is crystallographically equivalent. In the monoclinic distorted structures, two structurally unique metal sites, Ga1 and Ga2 (or In1 and In2), are present. Both maintain an octahedral environment, but they have different octahedral connections. This slightly different coordination environment for *m*-Ga-TBAPy and *m*-In-TBAPy originates from rotations of the benzoate groups due to stress during synthesis and activation,

a

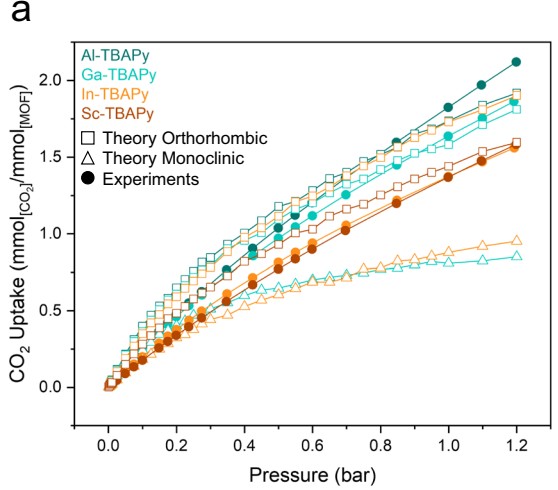

b

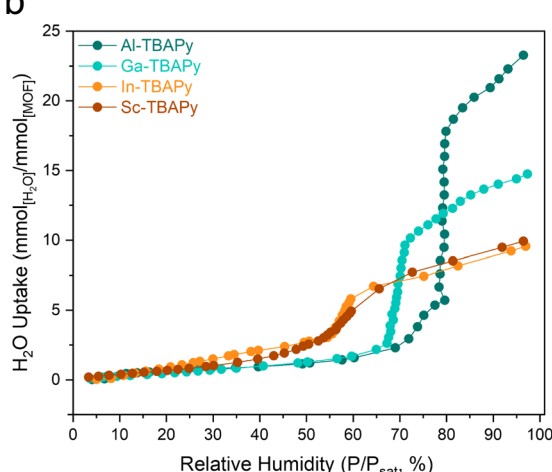

**Fig. 4 | CO₂ and H₂O Uptakes. a** CO₂ and (**b**) H₂O adsorption isotherms at 40 °C of the M-TBAPy structures (with M = Al, Ga, In, and Sc). Experimental isotherms at 40 °C (filled circles). Predicted isotherms of the orthorhombic structures (empty squares), predicted isotherms of the monoclinic structures (empty triangles). Source data are provided as a Source Data file.

respectively, which then result in a break-up of the linear arrangement of the metal ions.

### Characterization of the bulk material

The pore volumes of the materials are evaluated by carrying out $N_2$ adsorption isotherms at 77 K (Supplementary Note 7), and all materials demonstrate permanent microporosity, as shown by the type I $N_2$ isotherms[26] (Fig. S6a). The experimental pore volumes of the MOFs are compared with computational predictions, which generally tend to slightly over-predict the experimental results (Fig. S6b). In fact, computed pore volumes are based on perfect, fully activated, and infinitely large crystal models, thus showing higher values compared to experimental data[27]. The experimental and computational pore volumes of Al- and Sc-TBAPy are in good agreement. For In- and Ga-TBAPy, we report the computational pore volumes obtained for both the orthorhombic and monoclinic structures, with the experimental pore volume falling between those values.

The stability of the structures is evaluated via thermogravimetric analysis (TGA), which shows that all M-TBAPy MOFs present solvent loss up to 200 °C, and their structures are maintained until approximately 400–500 °C, at which point the decomposition of the TBAPy ligand starts to take place (Supplementary Note 8). The larger the effective ionic radius of the metal with oxidation state 3+ (i.e., $r_{Al^{3+}} < r_{Ga^{3+}} < r_{Sc^{3+}} < r_{In^{3+}}$)[28,29], the lower the stability of the MOF (Supplementary Note 8)[30,31]. The surface charge density of the metal center decreases from $[Al(H_2O)_6]^{3+}$ to $[In(H_2O)_6]^{3+}$, resulting in increased lability and higher ligand exchange rate for In-based complexes[32,33]. This is further reflected in the crystallization of the MOFs. Due to an increased tendency for ligand exchange in In(III) complexes, larger crystals of In-MOFs are more frequently grown, allowing for their characterization through single-crystal XRD (SC-XRD), as demonstrated in the original publication of In-TBAPy[22]. In contrast, Al- and Ga-MOFs are typically obtained as microcrystalline powders (Supplementary Note 9), and their structures are primarily determined using PXRD data[30], which also explains the lack of a single-crystal for Ga-TBAPy despite numerous synthesis efforts.

### CO₂, N₂, and H₂O adsorption isotherms

To study the adsorption behavior of the M-TBAPy MOFs at finite pressures and address the influence of the pyrene stacking distance

and metals on the MOFs' performance, the $CO_2$ adsorption isotherms of the different structures are determined experimentally and computationally (Fig. 4a). Here, we present the uptakes in $mmol_{[Adsorbate]}/mmol_{[MOF]}$ to emphasize that our results are not due to the weight of the metal incorporated in the structure. The factors used for conversion from mmol/g to mmol/mmol can be found in Supplementary Note 10. In the Henry regime, the uptakes of the simulated orthorhombic structures follow the trend expected if one looks solely at the inter-aromatic spacing: Al-TBAPy > Ga-TBAPy > In-TBAPy > Sc-TBAPy. Experimentally, we see a similar ranking: Al-TBAPy ≈ Ga-TBAPy > In-TBAPy ≈ Sc-TBAPy. In the low-pressure regime, the experimental $CO_2$ isotherms of Ga- and In-TBAPy align more closely with the computational data calculated from the monoclinic CIFs (Supplementary Note 11), while at high pressures, they correspond more accurately to the orthorhombic data (Fig. 4a). In fact, the simulated monoclinic structures reach the maximum $CO_2$ loading at much lower pressures compared to their orthorhombic counterparts, which may be ascribed to their lower pore volume. This may suggest that upon $CO_2$ loading, the structures may distort from monoclinic to orthorhombic. The phase transitions upon guest loading have been carefully investigated in the original publication of In-TBAPy, for which re-immersing the activated MOF (i.e., monoclinic structure) in fresh DMF, leads to a reversible transformation back to its orthorhombic structure[22]. We believe a similar structural flexibility occurs as the In- and Ga-TBAPy MOFs are loaded with $CO_2$ (Supplementary Note 12).

To investigate the $CO_2/N_2$ separation performance of the different frameworks, $N_2$ adsorption isotherms are also determined at 40 °C (Supplementary Note 13). The $N_2$ uptake of the M-TBAPy MOFs is approximately ten times lower than the $CO_2$ uptake, which is a strong indication that these materials preferentially bind to $CO_2$ compared to $N_2$.

Pure water vapor adsorption isotherms at 40 °C are also collected experimentally (Fig. 4b). Due to the greater complexity and lower reliability of computational $H_2O$ adsorption isotherms compared to $CO_2$ and $N_2$[34], here, we will only focus on the experimental results. We observe that Al- and Ga-TBAPy start to adsorb $H_2O$ at higher relative humidity levels (65–80%) than In- and Sc-TBAPy (30–40%). The minimal $H_2O$ uptake up to 65–80% relative humidity indicates the good performance of Al- and Ga-TBAPy for practical applications like $CO_2$ capture from wet flue gases, while In- and Sc-TBAPy have a slightly more hydrophilic behavior.

## Binding sites and binding energies

Binding sites and interaction energies are explored at a single-molecule level to understand further the framework-$CO_2$ interactions of the orthorhombic and monoclinic structure models. Our binding site analysis on pure $CO_2$ streams reveals that $CO_2$ is primarily located between stacks, aligning parallel to the pyrene cores (i.e., site A highlighted in Fig. 2), which is the energetically most favored state for the orthorhombic frameworks (Fig. 5). Here, the dominant interaction arises from the connection between the pyrene core's dispersed π-aromatic system and the molecular quadrupole moment of $CO_2$[35]. The calculated interaction energies follow a similar trend as the experimental $CO_2$ isotherms in the low-pressure regime. Al-TBAPy, with the optimum inter-aromatic spacing and highest $CO_2$ uptake, has the strongest interaction energy with a single $CO_2$ molecule (i.e., −32.0 kJ/mol) (Table 1 and Supplementary Note 14), followed by m-Ga-TBAPy, m-In-TBAPy, and Sc-TBAPy with slightly lower binding energy values. In the case of the monoclinic frameworks, the spacing decreases as the aromatic ligand no longer stacks in a parallel fashion due to a shift and tilt of the pyrene cores. This arrangement obstructs the $CO_2$'s access to the binding site between the pyrene stacks. In the minimum energy configuration of $CO_2$ in m-In-TBAPy and m-Ga-TBAPy, $CO_2$ occupies a central position within pore C, surrounded by the aromatic structures of the ligands (Fig. 5). For m-Ga-TBAPy, the interaction energy for this configuration amounts to -30.9 kJ/mol, while for m-In-TBAPy, the energy amounts to -27.7 kJ/mol. It is important to compare the DFT minimum energy configurations of the monoclinic and orthorhombic structures of Ga-TBAPy and In-TBAPy. In fact, this difference amounts to -3.68 meV/atom, and -12.70 meV/atom for Ga-TBAPy and In-TBAPy, respectively, with the monoclinic phase having a lower minimum energy in both cases, which indicates its higher stability with regards to its orthorhombic counterpart. The similar energy levels of these

structures in their respective configurations may account for their facile structural rearrangements under different conditions, which may also explain the presence of different phases in the same sample. As it can be seen from the DFT binding energies in pores B and C, the $CO_2$ molecule binds less strongly in these sites than in the minimum energy configuration between the parallel pyrene stacks (Supplementary Note 14), which once more highlights the higher affinity of this molecule for site A.

The binding sites of $N_2$ and $H_2O$ molecules are also shown in Fig. 5. For the orthorhombic structures, the preferential binding site of $N_2$ corresponds to site A, between pyrene stacks, but closer to the metal rod if compared to $CO_2$, which preferentially occupies the central region of the pore. On the other hand, $H_2O$ has two preferential adsorption sites: site A, and pore B, in close contact with the metal rod of the structure due to hydrogen bond interactions between $H_2O$ and the −OH groups coordinated to the metal. It can be noticed that $CO_2$, $N_2$, and $H_2O$ molecules have slightly different preferential adsorption sites, making these materials promising candidates for carbon capture from wet flue gases. In the case of the monoclinic structures, $CO_2$ preferentially sits in pore C, along with $N_2$. Given the smaller distance between stacks in their monoclinic configuration, these adsorbate molecules no longer fit between ligands. Similar to the orthorhombic structures, $H_2O$ has a preferential binding site that differs from the other gases, as it can mainly be found in pore B along the metal rods. This is further confirmed by the DFT binding energy calculations, where we show that $H_2O$ has a much higher affinity for site B, closer to the metal rod (Supplementary Note 14). For $N_2$ and $H_2O$ adsorbing in site A, we also observe much weaker DFT binding energies compared to the one of $CO_2$ in its minimum energy configuration. This also highlights the lower affinity of these gases to adsorb in the site between pyrene ligands. On the other hand, $H_2O$ has much stronger

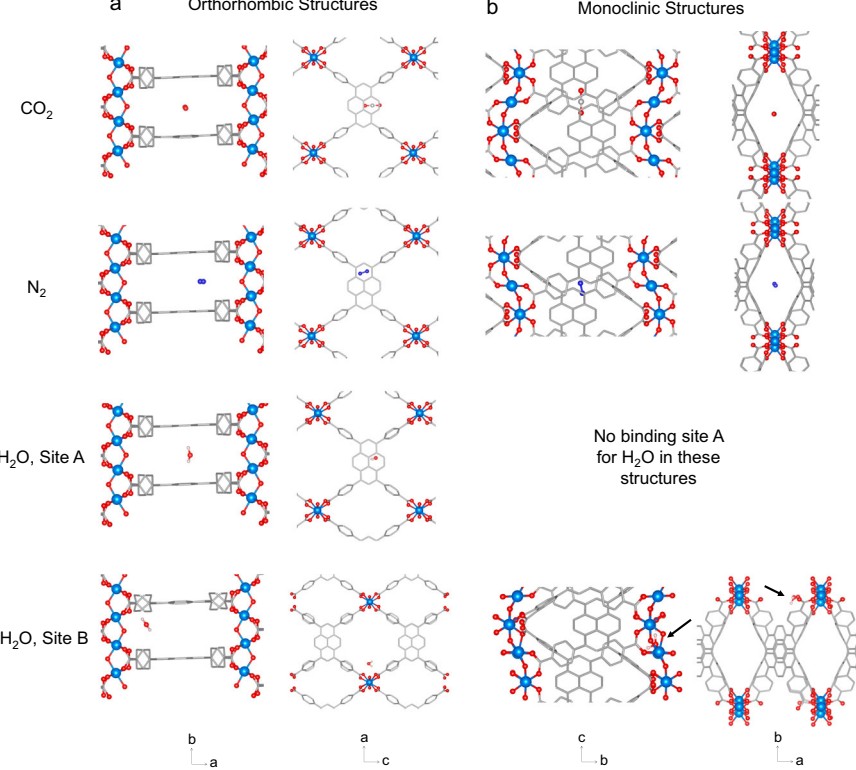

**Fig. 5 | Preferential $CO_2$, $N_2$ and $H_2O$ binding sites.** Most energetically favorable binding sites for (**a**) the orthorhombic, and (**b**) monoclinic structures. The monoclinic structures do not have a binding site A for $H_2O$, as the pyrene stacks are too narrow for water insertion. The images were generated from the corresponding

CIFs and cropped to emphasize the positions of the gas molecules within the structures. As a result, the periodicity of the MOFs may not be depicted. Color code: C (gray), O (red), metal (light blue), N (blue). Carbons (C) and hydrogens (H) are hidden for clarity apart from the $CO_2$ and $H_2O$ molecules.

                                              

interactions in site B for both the orthorhombic and monoclinic structures, showing its higher affinity for the site closer to the metal rod, as it favors hydrogen bond interactions.

To further assess the performance of these materials for carbon capture applications, we also report the experimental and computational isosteric heats of $CO_2$ adsorption ($Q_{st}$) (Supplementary Note 15). The experimental $Q_{st}$ is calculated using variable temperature adsorption isotherms at 25, 40, and 55 °C, and the values at zero loading follow a similar trend as the computational $Q_{st}$. These results are further supported by the DFT-calculated binding energies for the minimum energy configurations of the orthorhombic Al- and Sc-TBAPy and monoclinic structures of Ga- and In-TBAPy.

## Density maps
A graphical representation of the $CO_2$ center-of-mass (COM) probability distribution within the MOFs at 1 bar is depicted in Supplementary Note 16. For the structures in which the orthorhombic cell is maintained, and therefore the pyrene ligand stacks in a parallel manner along the *b*-axis, the $CO_2$ molecules are primarily located within stacks, with a marked preference for the central region. At 1 bar, the adsorbate density shows $CO_2$ first positioning itself centrally between the parallel pyrene stacks (site A), subsequently filling the elliptical pores B and C. In contrast, the distorted structures (i.e., *m*-In-TBAPy and *m*-Ga-TBAPy) consistently show reduced $CO_2$ density. The space between the pyrene stacks is too narrow for $CO_2$ to adsorb, and only one of the pores along the *c*-axis is accessible to $CO_2$, with the preferential adsorption site being located between two neighboring ligands, in line with the pyrene cores. The density maps at different temperatures do not show marked differences. However, one can notice that the histograms at lower temperatures are much narrower, highlighting the presence of one adsorption site only. At higher temperatures, one can notice the broadening of the peak and the emergence of a possible secondary adsorption site.

## Fine-tuning uptakes: a mixed-cation-TBAPy MOF
To further tailor the gas adsorption properties of this family of MOFs, we leverage chemical substitution at the metal center to exploit its impact on the inter-aromatic spacing and, thus, the MOFs' uptakes. To this end, we synthesized mixed-cation TBAPy-MOFs (i.e., $Al_xSc_y$-TBAPy). Since both versions of the pure Al- and Sc-TBAPy MOFs maintain the *Cmmm* structure, only this type of metal mixture is investigated with different molar ratios of Al/Sc precursors (i.e., 0.20/0.80, 0.50/0.50, and 0.80/0.20) (Supplementary Note 17). Scanning electron microscopy (SEM) images of the three MOFs highlight the change in size and shape of the crystals: at higher Al content, we observe thin platelets characteristic of the pure Al-TBAPy MOF, while the $Al_{0.20}Sc_{0.80}$-TBAPy MOF shows more rounded crystals, characteristic of pure Sc-TBAPy (Supplementary Note 18). The accurate Al/Sc ratios of the as-made materials are determined via energy dispersive X-ray spectroscopy in a scanning electron microscope (EDX-SEM), which confirms the expected ratios.

The lattice parameters of the MOFs are obtained through Le Bail fits, utilizing the *Cmmm* unit cell. While the ligands align along the *b*-axis, it is impossible to accurately determine this lattice parameter because it only scatters in the high 2*θ* range, and the Bragg peaks related to the 0h0 reflections have very weak intensities, making them unsuitable for fitting purposes. Nonetheless, we observe an overall increase of the unit cell parameter *a* as the Al content rises (Supplementary Note 17), which further confirms that changing the metal center has a clear effect on the crystal lattice parameters, which can thus be used to influence the material's performance for a given application.

To assess the impact of local structural modifications and non-parallel pyrene stacks caused by different Al and Sc arrangements, we generate in silico CIFs with different Al and Sc configurations (Supplementary Note 19). The obtained PXRD patterns of the simulated structures are compared to the experimental PXRDs of $Al_{0.50}Sc_{0.50}$-TBAPy (Fig. S18a). The different computational results do not show significant differences in diffraction data, and match relatively well the experimental $Al_{0.50}Sc_{0.50}$-TBAPy pattern. The $CO_2$ and $N_2$ uptakes at 40 °C, as well as $N_2$ isotherms at 77 K and pore volume measurements for the $Al_{0.50}Sc_{0.50}$-TBAPy MOF are also conducted (Figs. S6 and S18). The results follow the expected trend, and we see that the $CO_2$ uptake of the mixed-metal MOF lies between the pure Al- and Sc-TBAPy ones. The computed isotherms of $Al_{0.50}Sc_{0.50}$-TBAPy with different Al *vs* Sc arrangements slightly over-predict the experimentally measured isotherm (Fig. S18b, c), but show that overall, varying the arrangement of the Al and Sc metals in the structure, does not have a significant impact on the uptake of the materials. Finally, the experimental $H_2O$ vapor adsorption isotherm at 40 °C shows that this structure starts adsorbing $H_2O$ between approximately 50–60% relative humidity, which falls between the values of the pure Al- and Sc-TBAPy structures (Fig. S18d).

## From the lab to practical applications
In this work, we employ the PrISMa platform[36] to assess the performance of the synthesized orthorhombic pyrene-based MOFs for capturing $CO_2$ from a coal-fired power plant in the United Kingdom (UK), utilizing a Temperature Vacuum Swing Adsorption (TVSA) process at 0.6 bar. Here, the platform is used to comprehensively evaluate the materials for carbon capture by assessing their cost-effectiveness and scalability, which are crucial parameters for transitioning from laboratory experiments to pilot and demonstration projects. Moreover, we also use the platform to evaluate their environmental impact throughout the entire life cycle of the plant. This ensures that deploying these technologies results in a net reduction of $CO_2$-equivalent emissions.

Figure 6 provides a detailed comparative analysis of the performance of the various MOFs under dry and wet conditions. The subplots collectively illustrate the trade-offs among several key performance indicators (KPIs), including recovery, purity, productivity, specific thermal energy requirements, climate change impact, the use of natural resources such as minerals and metals (MR:MM), and net carbon avoidance cost (nCAC). This comprehensive visualization enables the identification of MOFs that provide the optimal balance of performance and cost-effectiveness for $CO_2$ capture. For a more detailed description of the case study and these KPIs, please refer to the work of Charalambous et al.[36].

Figure 6a illustrates the effectiveness and cost efficiency of different M-TBAPy MOFs in recovering $CO_2$. Al-TBAPy demonstrates a moderate recovery of 88% with a relatively low nCAC of 175€ $t^{-1}_{CO_2}$ under dry conditions, indicating a cost-effective solution with a decent recovery level. Ga-TBAPy shows a similar recovery performance to Al-TBAPy but with a slightly higher nCAC. In-TBAPy has a lower recovery rate and a higher nCAC, making it less effective and more expensive. Sc-TBAPy achieves the lowest recovery of 80% and the highest nCAC of 580€ $t^{-1}_{CO_2}$ under dry conditions. It is important to note that for this specific case study, the M-TBAPy MOFs remain less competitive than the monoethanolamine (MEA) benchmark, which has a nCAC of 100€ $t^{-1}_{CO_2}$ [36]. Figure 6b evaluates the purity of the captured $CO_2$ relative to the associated cost. The trend reflects that of recovery, with Al-TBAPy achieving the highest purity at 77% and Sc-TBAPy the lowest at 63% under dry conditions. This figure indicates that none of the evaluated materials meet the purity requirement for geological storage, which is set at 96%[36]. This highlights a significant limitation in the current performance of the M-TBAPy MOFs, emphasizing the need for further optimization to reach the required purity levels for effective carbon sequestration. Figure 6c examines the productivity of M-TBAPy MOFs against the specific thermal energy required. Similar to previous trends, Al-TBAPy stands out with the highest productivity and the lowest specific thermal energy consumption, indicating efficient $CO_2$ capture with reasonable energy usage. Productivity declines and

                                                                 

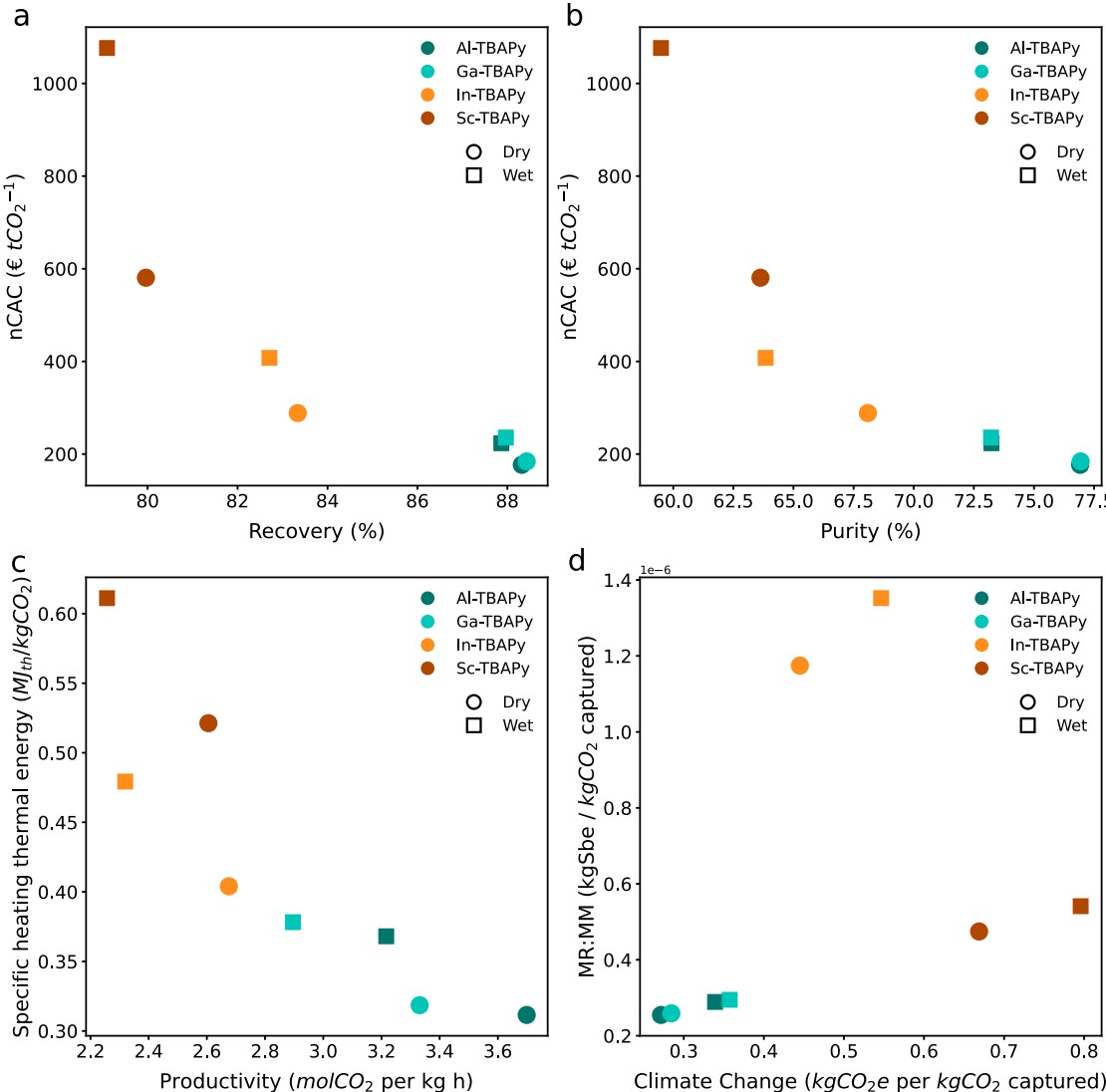

**Fig. 6 | M-TBAPy structures' performance in dry and wet conditions.** The structures were tested for a temperature-vacuum swing adsorption (TVSA) carbon-capture process at 0.6 bar added to a coal-fired power plant in the United Kingdom (UK) under dry and wet conditions. **a** The net carbon avoidance cost (nCAC) versus recovery. **b** The nCAC versus purity. **c** Specific thermal energy consumption for heating in MJ$_{th}$/kg versus productivity. **d** Minerals and metals (MR:MM) versus climate change. The nCAC describes the overall cost incurred in avoiding a mass unit of $CO_2$ into the atmosphere considering the total life cycle $CO_2$-eq. emissions of the capture plant in € $t_{CO_2}^{-1}$. Recovery quantifies the amount of $CO_2$ in the product stream divided by the total amount of $CO_2$ entering the column in %. Purity quantifies the amount of $CO_2$ in the product stream divided by the total amount of product in %. Productivity is obtained by dividing the working capacity in mass units by the full cycle time in mol kg$^{-1}$h$^{-1}$. Climate change indicates the total Global Warming Potential (GWP) due to greenhouse gas emissions to the air and $CO_2$ uptake from the atmosphere according to the Intergovernmental Panel on Climate Change (IPCC, 2013) in kg $CO_2$-Eq. MR:MM indicates the use of non-renewable nonfossil natural resources, e.g., minerals and metals in kg Sb-Eq, where Sb is antimony. Source data are provided as a Source Data file.

specific thermal energy consumption increases in the following order of metals: Ga, In, and Sc. Figure 6d addresses the environmental impacts of M-TBAPy MOFs, focusing on the balance between greenhouse gas emissions and the use of material resources (MR:MM). Ideal values are lower on both axes, indicating a smaller environmental footprint. Al and Ga MOFs exhibit the lowest climate change impact and MR:MM, highlighting their environmental efficiency. In-based MOFs show a moderate climate change impact, but are resource-intensive.

Figure 6 clearly demonstrates the impact that water has on the performance of the different MOFs studied here when introduced in the feed. In fact, the presence of water has a minimal effect on both the purity and working capacity across all MOF structures. Similarly, the recovery rates remain unaffected by the presence of water. The altered composition of the product stream, characterized by reduced $CO_2$ content and increased $H_2O$, results in more intensive energy

requirements for regenerating the MOFs, thereby causing an increase in the nCAC values. The extent to which different metals are influenced by water varies, largely due to the specific interactions between $H_2O$, $CO_2$, and the metal node. For instance, Sc-TBAPy is particularly sensitive, with its nCAC value nearly doubling in the presence of water, whereas Al-TBAPy exhibits a more moderate increase of 25% in its nCAC.

## Discussion
Many factors are impacted if we change the metal. The most prominent one is the effective ionic radius: $r_{Al^{3+}} < r_{Ga^{3+}} < r_{Sc^{3+}} < r_{In^{3+}}$ [29]. Such an increase in ionic radii lowers the spatial overlap between the oxygen orbitals of the carboxylate groups of the ligand and the metal, resulting in weaker bonds, larger metal-oxygen (M-O$_{Ligand}$) distances, and thus higher inter-aromatic spacing between ligands. However, although Sc(III) has a smaller ionic radius than In(III), its inter-aromatic spacing is

bigger. By looking at the M-O-M angle, we notice that In-TBAPy has a much smaller angle compared to the other MOFs, which consequently leads to the smaller inter-aromatic spacing observed in this structure.

In this work, we analyzed four pyrene-based MOFs (Al-, Sc-, In-, and Ga-TBAPy) for $CO_2$ capture. Our findings indicate that computational predictions may not fully account for all the factors necessary to identify the most efficient materials for practical $CO_2$ capture. Experimental evidence shows that at least two of the materials studied (i.e., In- and Ga-TBAPy) can exhibit different polymorphs, and phase transitions upon the removal of solvent molecules or the insertion of $CO_2$ may occur. For In-TBAPy, the structural change upon activation, and for Ga-TBAPy, a distinctive coordination pattern when using the same synthesis conditions as Al-, In- and Sc-TBAPy, result in different adsorption sites and smaller inter-aromatic spacing. To increase the accuracy of predictions, different phases and respective structural models should be considered. Moreover, we further explore the tunability of these MOFs by also investigating the incorporation of different cations (i.e., Al and Sc) within the same structure. Our results highlight the versatility of these materials, rendering them tailor-made adsorbents not only for $CO_2$ capture, where $\pi$ - electrostatic interactions play a crucial role, but potentially extending their applicability to other molecules, which may exhibit similar interactions.

When assessing the cost-effectiveness and scalability of these sorbents, we see that Al-TBAPy consistently performs well across multiple metrics, making it the most balanced and favorable MOF for $CO_2$ capture in this study. Ga-TBAPy is a close second, offering similar benefits with slightly higher costs. In-TBAPy and Sc-TBAPy are less desirable due to their lower recovery rates and higher costs and environmental impact. Although these physisorbent materials do not match the performance of currently available MEA technologies for practical $CO_2$ applications, in this study, we highlight their potential and insights, which can be useful for the development of new sorbent materials for $CO_2$ capture. Here, we emphasize how the MOF's chemistry (i.e., chemical composition, coordination chemistry), and activation can influence gas adsorption. For accurate computational predictions (often used in large screenings of datasets), it is crucial to account for these materials' changes (i.e., distortions, presence of additional phases) and their substantial consequence on the overall predictive outcomes.

## Methods

### Experimental Methods

**Powder x-ray diffraction (PXRD).** PXRD data on all samples were collected on a Bruker D8 Advance diffractometer at ambient temperature using monochromated Cu K$\alpha$ radiation ($\lambda$ = 1.5406 Å), with a $2\theta$ step of 0.02° with different $2\theta$ ranges. Simulated PXRD patterns were generated from the corresponding crystal structures using Mercury 3.0. Images of the structures were generated using Vesta (4.6.0). Synchrotron data for the activated materials were obtained with $\lambda$ = 0.9572 Å. Similarly, in-situ variable temperature PXRD under dynamic vacuum was performed for Ga-TBAPy with $\lambda$ = 0.69437 Å. Indexing and refinements were performed using the TOPAS-academic $v6$ software package[37].

**Micro-electron diffraction (microED/3D-ED).** MicroED/3D-ED experiments were performed on an XtaLAB Synergy-ED electron diffractometer consisting of a 200 kV emitter (corresponding to a wavelength of 0.0251 Å), lens system, HyPix-ED detector and controlled by CrysAlisPro for ED (Rigaku, V1.171.43.51a, 2022). The sample was spread on a TEM grid and then measured at room temperature and vacuum conditions. Data reduction, scaling, and absorption corrections were performed using CrysAlisPro (Rigaku, V1.171. 42.52a, 2022). A multi-scan absorption correction was performed using CrysAlisPro 1.171.42.52a (Rigaku, 2022) using spherical harmonics, implemented in

SCALE3 ABSPACK scaling algorithm. Using Olex2[38], the structure was solved with the SHELXT[39] structure solution program using Intrinsic Phasing and refined with the olex2.refine[40] refinement package using Gauss-Newton minimization. All non-hydrogen atoms were refined anisotropically. Hydrogen atom positions were calculated geometrically and refined using the riding model at distances derived from Neutron diffraction data.

**Thermal gravimetric analysis (TGA).** A PerkinElmer Thermogravimetry Analyzer was used to determine the decomposition temperature of the samples. All measurements were performed under airflow up to 700 °C.

**Scanning electron microscope (SEM).** The morphological characteristics were investigated by a Thermo Scientific Teneo SEM operated at an accelerating voltage that varied between 2 and 5 kV. For SEM measurements, all samples were deposited on a carbon tape.

**Energy dispersive x-ray spectroscopy (EDX).** These measurements were also conducted on a Thermo Scientific Teneo SEM instrument at 15 kV accelerating voltage and using a Bruker X-Flash 6-30 detector.

**Gas adsorption measurements.** The $N_2$ adsorption isotherm measurements at 77 K were performed by using BELSORP Mini (BEL Japan, Inc.). Before measurements, samples were activated at 170 °C for 12 h under a dynamic vacuum.

For gas adsorption measurements at 25, 40 and 55 °C, the samples were placed in Micromeritics adsorption cells and activated under vacuum (0.02 mbar), heated up to 170 °C and remained at the same temperature for 12 h using an activation station Micromeritics VacPrep 061. After activation, the samples were cooled down naturally to room temperature. Next, the evacuated cells containing degassed samples were transferred to a balance and weighed to determine the mass of the sample after activation. The adsorption cells were then transferred to the analysis ports of the instrument Micromeritics 3Flex, where $CO_2$ (99.998% gas purity) and $N_2$ (99.999% gas purity) adsorption isotherms measurements at different temperatures were performed using an isothermal water bath.

**Water vapor isotherms.** For all water vapor adsorption isotherms, the MOFs were activated under vacuum ($10^{-2}$ kPa) at 170 °C, for 12 h using an activation station Belsorp vac II. The adsorption cell was then transferred to the analysis port of the instrument Belsorp aqua, where water vapor adsorption isotherms at 40 °C were measured.

**Nuclear magnetic resonance (NMR).** A Bruker Avance III 400 MHz spectrometer equipped with a 5mm BBFO Z-gradient SmartProbe was used. $^1$H-NMR spectrum was acquired using standard pulse sequences from the Bruker library. Data was processed using the Mestrenova software.

### MOFs syntheses

**Syntheses of Al-TBAPy[12], In-TBAPy[22], Sc-TBAPy[23], and Ga-TBAPy.** In a 12 mL glass vial, a mixture of either Al(NO$_3$)$_3$ · 9 H$_2$O, In(NO$_3$)$_3$ · x H$_2$O, Sc(NO$_3$)$_3$ · x H$_2$O or Ga(NO$_3$)$_3$ · x H$_2$O (0.03 mmol) and TBAPy (0.02 mmol, 10 mg) were introduced into a solvent mixture of DMF/dioxane/H$_2$O (4 mL, ratio 2/1/1). Concentrated HCl (32 wt%) (10 $\mu$L) was added. The vials were sonicated and then heated to 85 °C for 12 h, with heating and cooling rates of 2 and 0.1 °C/min, respectively. The solids were recovered by centrifugation and washed with DMF three times. Before adsorption measurements, solvent exchange with either acetone or ethanol was performed overnight. The MOFs were then activated at 170 °C for 12 h under dynamic vacuum. Yields: Al-TBAPy: 61%, Ga-TBAPy: 45%, Sc-TBAPy: 61%, and In-TBAPy: 66%.

**Synthesis of Al$_x$Sc$_y$-TBAPy.** (with molar ratios of $x/y$ = 0.80/0.20, 0.50/0.50, and 0.20/0.80). The syntheses of these MOFs follow the same procedure as stated above for the remaining TBAPy structures, but instead of single metal precursors, a mixture of both Al(NO$_3$)$_3$ · 9 H$_2$O and Sc(NO$_3$)$_3$ · x H$_2$O with the corresponding amounts of metal precursors: 0.80/0.20 (0.024 mmol - 0.006 mmol), 0.50/0.50 (0.015 mmol−0.015 mmol) or 0.20/0.80 (0.006 mmol−0.024 mmol) were added. The MOFs were washed with DMF three times. Before adsorption measurements, solvent exchange with either acetone or ethanol was performed overnight. Yields: Al$_{0.20}$Sc$_{0.80}$-TBAPy: 40%, Al$_{0.50}$Sc$_{0.50}$-TBAPy: 60%, Al$_{0.80}$Sc$_{0.20}$-TBAPy: 41%.

## Computational methods
In this study, the DFT calculations were conducted using the Quickstep code of the CP2K package (version 9.1)[41]. This code is an efficient DFT implementation for large and complex structures by exploiting the mixed Gaussian and plane waves (GPW) method alongside pseudopotentials, optimizing the wave function with the orbital transformation (OT) technique. We employed double-zeta DZVP-MOLOPT-SR contracted basis sets and GTH pseudopotentials to represent the electronic wave function. The multigrid used for plane waves had a 4-level structure with a primary cutoff of 600 Ry, a relative cutoff of 50 Ry, and a progression factor of 3. For the exchange-correlation energy, we applied the Perdew-Burke-Ernzerhof (PBE)[42] functional with the DFT-D3(BJ)[43] model to account for many-body dispersion interactions. The selection of integration grid and DFT functional was based on the research of Ongari et al.[44], who provided optimized DFT settings for accurate and efficient high-throughput computational analysis of covalent organic frameworks (COFs) and MOFs.

The RASPA[45] was used to perform MC simulations. The optimized framework geometries were kept rigid in all classical simulations. We considered van der Waals and electrostatic interactions to describe the energy surface, represented respectively by the Lennard-Jones (LJ) potential and Coulomb interactions. Periodic boundary conditions were employed with a cutoff radius of 12.8 Å, including tail corrections to remedy the truncation. Density-derived electrostatic and chemical (DDEC) method is used to compute the partial charges on the atoms of the MOF frameworks[46]. The Ewald summation technique was used to model Coulomb interaction[47]. The TraPPE force field[48] was selected to model gas-gas interactions for CO$_2$ and N$_2$. The dispersion interactions of the framework and the gases were modeled with Lennard-Jones potentials. The Lennard-Jones parameters are refined from the Universal Force Field (UFF)[49] according to the experimental CO$_2$ adsorption isotherm[50].

Unified workflows were used for several parts of this study to ensure the reproducibility and direct comparability of computed data. The Automated Interactive Infrastructure and Database for Computational Science (AiiDA)[51] was employed to orchestrate the different steps, managing the interaction of different codes and providing automation and similarity of the calculations. The workflows are published and maintained as the *aiida-lsmo* plugin on GitHub[52].

**Structure optimization workflow.** First, a preliminary cell optimization step was performed for a maximum of 200 cycles to optimize the frameworks, keeping the unit cell angles fixed. This step ensures that the atom positions were relaxed and an adequate minimum was found based on the experimental cell parameters before optimizing the unit cell vectors. As a result, the simulated cell was kept closer to experiments. Subsequently, a final cell optimization was performed without constraints on the unit cell's angles. In the first step, the Broyden-Fletcher-Goldfarb-Shanno (BFGS) minimizer is used, and the limited-memory BFGS minimizer is employed for the final cell optimization. The threshold for the pressure was set to 100 bar. The system was considered converged when the maximum forces on the atoms dropped below $0.45\,\mathrm{m}E_\mathrm{h}\,a_0^{-1}$ and the geometry change between the

current and the last optimizer iteration was lower than $3\,\mathrm{m}a_0$, according to the default settings implemented in CP2K[53]. Because of the large quadrupole moment of CO$_2$, it is important to have an accurate model of atomic point charges to evaluate the electrostatic interactions in classical simulations. The Density Derived Electrostatic and Chemical (DDEC) approach is exploited to assign net atomic charges (NACs) to the framework atoms of the optimized structures. In this work, the DDEC6 method, as implemented in the Chargemol software[54], was used to perform atomic population analysis based on the electron density obtained from a DFT calculation using the settings as detailed above.

Geometric properties of the materials (pore volumes and pore diameters) were evaluated using the software Zeo++[55]. Pore volumes were accessed using the probe-occupiable pore volume model as implemented in the software. This technique provides a computational pore volume definition directly related to experimental pore volumes obtained from nitrogen isotherms[27].

**Single-component isotherm workflow.** The *Isotherm* work chain in the *aiida-lsmo* plugin was used to generate pure CO$_2$, and N$_2$ adsorption isotherms. The adsorption isotherms were simulated in the Grand Canonical ensemble (GCMC ensemble)[56]. Here, 15,000 cycles were used for equilibration and 15,000 cycles for production. Simulations at subsequent pressure points were performed starting from the restart file of the previous pressure step, thus reducing the number of cycles necessary for initialization.

**Binding site workflow.** The minimum energy configuration of adsorbate molecules (CO$_2$, N$_2$, H$_2$O) in the different frameworks was determined using the Monte Carlo (MC) simulations. The force field (FF) parameters were kept the same as mentioned above, and $10^4$ MC cycles were used to determine the molecule position. Combined with density maps, we recorded the energies from FF simulations and subsequently optimized those with adsorbate molecules inserted configurations with minimal energies in DFT simulation. Both MOF structures and adsorbate molecules are fully relaxed. Then we performed basis set superposition error (BSSE) correction simulation[57] to cancel the energy difference resulting from basis functions overlap. The adsorption binding energy can be calculated using Eq. (1):

$$E_{\mathrm{BD}} = E_{\mathrm{system}} - E_{\mathrm{MOF}} - E_{\mathrm{adsorbate}} \tag{1}$$

where $E_{\mathrm{system}}$ is the BSSE corrected energy of the relaxed MOF and adsorbates, and $E_{\mathrm{system}}$ is the energy of the relaxed MOF structure without an adsorbate.

## Data availability
The data generated in this study have been deposited in the Zenodo database (https://doi.org/10.5281/zenodo.14387276)[58]. Source data are also provided with this paper. Source data are provided with this paper.

## Code availability
The codes used in this study can be found in the work of Charalambous et al.[36], Li et al.[50], as well as this reference[59].

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

## Acknowledgements

The authors acknowledge Dr. Dmitry Chernyshov and Dr. Wouter VanBeek (European Synchrotron Radiation Facility, Grenoble, France) for contributing to this work. The authors also acknowledge Sho Ito (Rigaku Corporation, Japan) for microED/3D-ED data collection. N.P.D. would also like to thank Sanjay Venkatachalam and Pratap Narayan Soni for their help with the instruments and training, and Balázs Novotny for advice and scientific discussions. This work was supported by the MARVEL National Centre for Competence in Research funded by the Swiss National Science Foundation (grant agreement ID 51NF40-182892). The authors acknowledge PRACE and MARVEL for awarding access to Piz Daint (project ID: pr128) and Eiger (project ID: mr30) at the Swiss National Supercomputing Centre (CSCS), Switzerland.

## Author contributions

N.P.D., F.P.U., and C.P.I. conceived and designed the project. N.P.D. performed most of the experiments, and N.P.D., F.P.U., and C.P.I. analyzed the experimental data. N.P.D. and J.E. measured $CO_2$, $N_2$ and $H_2O$ isotherms of all MOFs and prepared samples for synchrotron experiments. W.L.Q. provided some of the equipment and helpful feedback. M.J.P., Y.L., E.M., and X.J. performed the simulations, and analyzed the computational data with the help of A.O-G.. C.S. solved the structure of Ga-TBAPy via microED/3D-ED measurements. P.S. analyzed and refined the synchrotron X-ray diffraction data. E.O. performed SEM-EDX imaging and quantification. N.P.D., F.M.E., and B.S. wrote the manuscript with the computational inputs of M.J.P., Y.L., E.M., and X.J. All authors reviewed, edited, and approved the manuscript.

## Competing interests

The authors declare no competing interests.
