## [Transparent Peer Review file · Nature Communications]

Unraveling Metal Effects on CO₂ Uptake in Pyrene-based Metal-Organic Frameworks

Corresponding Author: Professor Berend Smit

Version 0:

Reviewer comments:

Reviewer #1

(Remarks to the Author)

The authors investigated metal effects on the CO₂ capture performance and structural properties in orthorhombic M-TBAPy (M = Al, Ga, In, Sc) MOFs using experimental and computational approaches. The authors also found that the newly-identified monoclinic phase and the ground state orthorhombic phase can coexist in Ga- and In-TBAPy MOFs. I think the detailed atomic-scale mechanism is well explained, and the results are convincing. The manuscript is also clearly written. Therefore I highly recommend publication after the authors address the following issues.

1. I'm curious about the total energy difference between the orthorhombic and monoclinic phases in Ga- and In-TBAPy MOFs. This could explain the coexistence of two distinct phases.

2. In Fig. 2 and 5, the authors show the crystal structures and CO₂ distribution. But, it's also nice if the authors consider to show geometries of CO₂ adsorption structures with zoomed-in images.

Reviewer #2

(Remarks to the Author)

The currently submitted paper presents a structural, computational, and gas-adsorption study of four metal-organic framework (MOF) materials: Al-, Sc-, Ga-, and In-TBAPy. This is a follow-up study of some of the authors' previous excellent work in Nature 2019 (10.1038/s41586-019-1798-7). The currently submitted manuscript, however, does not intellectually progress beyond its predecessor. Neither of the presented materials is newly synthesized and the conclusions reached now were accurately predicted in Nature 2019:

1. Nature 2019 predicts (10.1038/s41586-019-1798-7, Figure 1c) and confirms (10.1038/s41586-019-1798-7, Figure S1.10a) the preferential CO₂ adsorption site in Al-TBAPy in between of parallelly stacked pyrene cores. Therefore, the presence of such a site in Al-TBAPy (the same material!) and the isostructural materials Sc-, Ga-, and In-TBAPy (Figure 5) is not revelatory.

2. Nature 2019 optimizes the structures of Al-, Sc-, Ga-, and In-TBAPy with DFT (10.1038/s41586-019-1798-7, Extended Data Table 2) and postulates a relationship between CO₂ uptake and pyrene-pyrene stacking distance (10.1038/s41586-019-1798-7, Extended Data Figure 5). The current study merely tests this prediction (Figure 4) and reaches the same conclusions.

3. Nature 2019 explores the CO₂ binding sites using Rietveld refinement, discusses the CO₂/N₂ separation from simulated flue gases, and investigates the influence of humidity on CO₂ adsorption. These important aspects of carbon capture were not considered at all in the current manuscript.

4. The topic taken up by the authors (metal effects on CO₂ uptake in MOFs) is not novel since the first report of such influence dates back to 2008 (10.1021/ja8036096).

The currently submitted incremental and unoriginal study does not meet the standards of "Nature Communications" and shall not be published.

Besides the uncorrectable lack of novelty, the study also presents deplorably sub-standard technical quality:

5. The aromaticity scheme shown in Figure 1 and Figure S1 is incorrect.

6. The scheme shown in Figure S1 does not match the description of the Synthesis of 1 (both on page S3). The synthesis of

- 1 and 1' is described in part using a non-standard imperative instead of past tense.
7. Yields of MOF materials syntheses are not reported.
8. The statement claiming that Ga-TBAPy has not been reported is incorrect: 10.1039/D3TA05297D.
9. The discovery of the monoclinic phase of Ga-TBAPy from electron diffraction is interesting but not given much attention. The space group is inconclusively determined as P2(1)/n or P2/c in Supplementary note 5.
10. Rietveld refinement of the powder XRD pattern of Ga-TBAPy accounting for both the orthorhombic and the monoclinic phases (Figure S7) is not satisfactory as there is still a clear difference between the model and the data, which the authors explicitly admit ("data quality does not allow us to refine the actual structure"). The comparison between the calculated and experimental patterns (Figure S8 and Figure 3d) is equally inconclusive. Therefore, assertions about Ga-TBAPy ("structural distortion", "structural rearrangement", "lattice distortion", "Ga-TBAPy [...] no longer shows the perfect stacking of the TBAPy ligand", "rotations of the benzoate groups due to stress during synthesis and activation") may be inaccurate. Presence of phases other than o-Ga-TBAPy and m-Ga-TBAPy should equally not be excluded.
11. The hkl tags are incorrect in all Rietveld refinements (Figures S4-S7).
12. The interpretation of the 77 K N₂ isotherm of Al-TBAPy (Figure S10a) as type II is incorrect.
13. The claim that Al(0.50)Sc(0.50)-TBAPy features a scheme of alternating Al and Sc atoms in its structure (Figure S3a) is dubious. Any other arrangement of these atoms may equally be possible.
14. In Figure 4, CO₂ adsorption isotherms were plotted on an unconventional log-log scale. Moreover, the x-axes should be labeled as "Pressure" and not "log(Pressure)" since $\log(0.01) = -2$, $\log(0.1) = -1$, and $\log(1) = 0$. The y-axes are equally mislabeled.
15. In the discussion of Figure 4, the experimental CO₂ uptake of Ga-TBAPy is thought to lie in between of the trends computed for o-Ga-TBAPy and m-Ga-TBAPy. However, experimental CO₂ uptake of Al-TBAPy is not accurately computed neither, despite the MOF occurring in its orthorhombic phase only.
16. The authors discuss the metal effects on CO₂ uptake in pyrene-based MOFs without considering the density of the studied materials. It is striking that the MOF constructed from the lightest metal element (Al) displayed the highest uptake [expressed in mmol(CO₂)/g(MOF)], while the MOF constructed from the heaviest metal element (In) displayed the lowest uptake. An explanation that different observed uptakes are just the consequence of different MOF densities and not different adsorbent-adsorbate interactions was not considered.
17. Experimental values of isosteric heats of CO₂ adsorption, now standard in the literature on carbon capture, are not determined.
18. In the discussion of TGA as well as that of atomic (or ionic, the text confuses the two) radii, Sc-TBAPy is somehow omitted as if it had not made part of the series. The inclusion of Al(0.50)Sc(0.50)-TBAPy at the end of the manuscript also seems as an afterthought.

Reviewer #3

(Remarks to the Author)

This manuscript concerns design of better CO₂ capture physisorbents. In prior work, a design paradigm had been proposed that spaces pyrene linkers in MOFs at roughly 0.7 nm to create pockets favorable for CO₂. This work studies such systems with four metal variations that allow for effects of small changes in spacing to be studied with regards to CO₂ separations. The authors report mixed phases in some examples showing a more complex scenario exists than predicted theoretically. Experimental CO₂ binding is studied and the binding sites extracted computationally. I feel the work is generally well-executed but there is a main point I feel that needs to be addressed. The entire point of the pyrene paradigm is that the binding sites in such systems would allow physisorptive CO₂ binding in wet gases. This point is mentioned in the introduction but not addressed in the experiments. There are numerous studies on subtle structural variation of structure by metal exchange in MOFs including for CO₂ capture. Given the paradigm has already been presented for this exact family, I do not feel that metal variation and its effects is itself sufficient novelty to merit such a high impact journal. If the authors can address the water co-sorption, this can be reconsidered. It is still suitable for a high impact chemistry-specific journal.

Version 1:

Reviewer comments:

Reviewer #1

(Remarks to the Author)

I'm OK with it being published in the current form.

Reviewer #2

(Remarks to the Author)

The authors have successfully addressed all the comments raised during the review. The intensity and fiery nature of the review process underscore the importance of this work. We recommend the manuscript for publication.

Reviewer #3

(Remarks to the Author)

My prior comment related to the critical absence of CO₂/water co-sorption data. The authors have added water sorption. They have not experimentally addressed the point of water/CO₂ co-adsorption. This is not an exceptional request for a work

making the claims in this work. The use of the Prisma platform to extract process level metrics for “wet” systems is not a substitute for water/CO₂ breakthrough experiments. In fact, to extrapolate process metrics when a crucial experiment has not been done sets a poor precedent. Given that the family of compounds has been reported in Nature previously, I do not feel this work is a significant improvement in terms of carbon capture, certainly not at the level to merit such a high profile venue. If this was for JACS or Angewandte Chemie, I would still expect to see the dynamic column breakthrough in wet gas as the claim is central to the novelty and impact of this work.

Version 2:

Reviewer comments:

Reviewer #3

(Remarks to the Author)

This is a detailed study in an important area. The notion that metal ion variation in MOFs can be used to tweak pore structure and consequently gas separation performance is well established. I give a small sample of examples (Ultrasensitive sorption behavior of isostructural lanthanide-organic frameworks induced by lanthanide contraction, JOURNAL OF MATERIALS CHEMISTRY 2012 10.1039/c2jm33884j; Rationally tuning the separation performances of [M₃(HCOO)₆] frameworks for CH₄/N₂ mixtures via metal substitution, MICROPOROUS AND MESOPOROUS MATERIALS 2016, DOI10.1016/j.micromeso.2016.01.030; Lanthanide contraction effects on the structures, thermostabilities, and CO₂ adsorption and separation behaviors of isostructural lanthanide-organic frameworks, CRYSTENGCOMM 2015 10.1039/c4ce02073a; Reticular Chemistry in Pore Engineering of a Y-Based Metal-Organic Framework for Xenon/Krypton Separation, ACS APPLIED MATERIALS & INTERFACES 2023 10.1021/acscami.3c01229)

The fact in this work that the peripheral phenyl ring rotation is also impacted and not in a way that necessarily enhances performance really does not strike me as sufficient novelty and impact for a journal of this level especially considering already published work by this group. I defer to the editors though.

Response to the Reviewers

The authors would like to thank the editor and reviewers for their valuable comments and suggestions, which significantly enhanced the quality and clarity of the manuscript. The manuscript has been revised according to the editor and reviewers' feedback. Below, you will find our detailed responses. Figures and tables have only been added to the revised version of the manuscript and supporting information to make this point-by-point response more concise. For the editor's and reviewers' convenience, we have highlighted significant changes in the revised version in blue.

Additional Data Added to the Manuscript

We experimentally investigated the pure H₂O adsorption of our materials. We started by collecting pure water vapor adsorption isotherms at 40 °C. We have updated the revised version of the manuscript accordingly:

Pure water vapor adsorption isotherms at 40 °C are also collected experimentally (Figure 4 b). Due to the greater complexity and lower reliability of computational H₂O adsorption isotherms compared to CO₂ and N₂ [1], here, we will only focus on the experimental results. We observe that Al- and Ga-TBAPy start to adsorb H₂O at higher relative humidity levels (65-80%) than In- and Sc-TBAPy (30-40%). The minimal H₂O uptake up to 65-80% relative humidity indicates the good performance of Al- and Ga-TBAPy for practical applications like CO₂ capture from wet-flue gases, while In- and Sc-TBAPy have a slightly more hydrophilic behavior.

And in the *Methods* section we have:

Water Vapor Isotherms. For all water vapor adsorption isotherms, the MOFs were activated under vacuum (10^{-2} kPa) at 170 °C, for 12 hours using an activation station Belsorp vac II. The adsorption cell was then transferred to the analysis port of the instrument Belsorp aqua, where water vapor adsorption isotherms at 40 °C were measured.

Reviewer 1

The authors investigated metal effects on the CO₂ capture performance and structural properties in orthorhombic M-TBAPy (M = Al, Ga, In, Sc) MOFs using experimental and computational approaches. The authors also found that the newly-identified monoclinic phase and the ground state orthorhombic phase can coexist in Ga- and In-TBAPy MOFs. I think the detailed atomic-scale mechanism is well explained, and the results are convincing. The manuscript is also clearly written. Therefore I highly recommend publication after the authors address the following issues.

Comment 1.1 — I'm curious about the total energy difference between the orthorhombic and monoclinic phases in Ga- and In-TBAPy MOFs. This could explain the coexistence of two distinct phases.

Reply: We have added a short paragraph in the main text describing these energy differences:

It is interesting to compare the DFT minimum energy configurations of the monoclinic and orthorhombic structures of Ga-TBAPy and In-TBAPy. In fact, this difference amounts to -3.68 meV/atom, and -12.70 meV/atom for Ga-TBAPy and In-TBAPy, respectively, with the monoclinic phase having a lower minimum energy in both cases, which indicates its higher stability with regards to its orthorhombic counterpart. The similar energy levels of these structures in their respective configurations may account for their facile structural rearrangements under different conditions, which may also explain the presence of different phases in the same sample. As it can be seen from the DFT binding energies in pores B and C, the CO₂ molecule binds less strongly in these sites than in the minimum energy configuration between the parallel pyrene stacks (Supplementary Note 14), which once more highlights the higher affinity of this molecule for site A.

Comment 1.2 — In Fig. 2 and 5, the authors show the crystal structures and CO₂ distribution. But, it's also nice if the authors consider to show geometries of CO₂ adsorption structures with zoomed-in images (Figure 5).

Reply: We have updated the manuscript with zoomed-in images of the binding sites for CO₂, N₂, and H₂O molecules in the structures. The revised version now reads:

The binding sites of N₂ and H₂O molecules are also shown in (Figure 5). For the orthorhombic structures, the preferential binding site of N₂ corresponds to site A, between pyrene stacks, but closer to the metal rod if compared to CO₂, which preferentially occupies the central region of the pore. On the other hand, H₂O has two preferential adsorption sites: site A, and pore B, in close contact with the metal rod of the structure due to hydrogen bond interactions between H₂O and the -OH groups coordinated to the metal. It can be noticed that CO₂, N₂, and H₂O molecules have slightly different preferential adsorption sites, making these materials interesting candidates for carbon capture from wet-flue gases. In the case of the monoclinic structures, CO₂ preferentially sits in channel C, along with N₂. Given the smaller distance between stacks in their monoclinic configuration, these adsorbate molecules no longer fit between ligands. Similar to the orthorhombic structures, H₂O has a preferential binding site that differs from the other gases, as it can mainly be found in pore B along the metal rods. This is further confirmed by the DFT binding energy calculations, where we show that H₂O has a much higher affinity for site B, closer to the metal rod (Supplementary Note 14). For N₂ and H₂O adsorbing in site A, we also observe much weaker DFT binding energies compared to the one of CO₂ in its minimum energy configuration. This also highlights the lower affinity of these

gases to adsorb in the site between pyrene ligands. On the other hand, H₂O has much stronger interactions in site B for both the orthorhombic and monoclinic structures, showing its higher affinity for the site closer to the metal rod, as it favors hydrogen bond interactions.

Reviewer 2

The currently submitted paper presents a structural, computational, and gas-adsorption study of four metal-organic framework (MOF) materials: Al-, Sc-, Ga-, and In-TBAPy. This is a follow-up study of some of the authors' previous excellent work in Nature 2019 (10.1038/s41586-019-1798-7). The currently submitted manuscript, however, does not intellectually progress beyond its predecessor. Neither of the presented materials is newly synthesized and the conclusions reached now were accurately predicted in Nature 2019:

Reply: We respectfully disagree with the reviewer's assertion that the results were accurately predicted in our 2019 article. In fact, in our 2019 article, we simply assumed that the distance between the pyrene rings would scale with the diameter of the metal. In this work, we show that this assumption was too simple because of the phase transitions, and the reality is much more complex and interesting than we expected.

Comment 2.1 — Nature 2019 predicts (10.1038/s41586-019-1798-7, Figure 1c) and confirms (10.1038/s41586-019-1798-7, Figure S1.10a) the preferential CO₂ adsorption site in Al-TBAPy in between of parallelly stacked pyrene cores. Therefore, the presence of such a site in Al-TBAPy (the same material!) and the isostructural materials Sc-, Ga-, and In-TBAPy (Figure 5) is not revelatory.

Reply: The importance of this work is not to show the CO₂ binding site in M-TBAPy MOFs. As we mentioned in our reply to the previous point, Boyd et al. [2] assumed all M-TBAPy materials are isostructural, and the significant outcome of our study is to highlight and demonstrate that this assumption is too simple.

We realize that this may not have been sufficiently clear in the original manuscript, and to avoid confusion on this point, we have added the following to the *Introduction*:

Moreover, given the possible different phases present in some of the MOFs studied (i.e., monoclinic and orthorhombic), we also demonstrate the effect that each of them has on the adsorption behavior of the materials, which challenges the predictions made by Boyd et al. [2]. While their computational study provides valuable insights for future design of sorbent materials, here we aim to highlight the importance of experimental validation, as small variations in the crystalline phase can lead to considerably different adsorption characteristics.

Comment 2.2 — Nature 2019 optimizes the structures of Al-, Sc-, Ga-, and In-TBAPy with DFT (10.1038/s41586-019-1798-7, Extended Data Table 2) and postulates a relationship between CO₂ uptake and pyrene-pyrene stacking distance (10.1038/s41586-019-1798-7, Extended Data Figure 5). The current study merely tests this prediction (Figure 4) and reaches the same conclusions.

Reply: Please, see points 2.0 and 2.1.

Comment 2.3 — Nature 2019 explores the CO₂ binding sites using Rietveld refinement, discusses the CO₂/N₂ separation from simulated flue gases, and investigates the influence of humidity on CO₂ adsorption. These important aspects of carbon capture were not considered at all in the current manuscript.

Reply: While the binding sites for CO₂ had been already determined in the previous version of our manuscript, here, we now also report the N₂ and H₂O binding sites (please, see comment 1.2).

The effect of humidity on the performance of these MOFs has been tested, and we have added an additional section (i.e., *From the Lab to Practical Applications*) to the manuscript:

From the Lab to Practical Applications. In this work, we employ the PrISMa platform [3] to assess the performance of the synthesized orthorhombic pyrene-based MOFs for capturing CO₂ from a coal-fired power plant in the United Kingdom (UK), utilizing a Temperature Vacuum Swing Adsorption (TVSA) process at 0.6 bar. Here, the platform is used to comprehensively evaluate the materials for carbon capture by assessing their cost-effectiveness and scalability, which are crucial parameters for transitioning from laboratory experiments to pilot and demonstration projects. Moreover, we also use the platform to evaluate their environmental impact throughout the entire life cycle of the plant. This ensures that deploying these technologies results in a net reduction of CO₂-equivalent emissions.

Figure 6 provides a detailed comparative analysis of the performance of the various MOFs under dry and wet conditions. The subplots collectively illustrate the trade-offs among several key performance indicators (KPIs), including recovery, purity, productivity, specific thermal energy requirements, climate change impact, the use of natural resources such as minerals and metals (MR:MM), and net carbon avoidance cost (nCAC). This comprehensive visualization enables the identification of MOFs that provide the optimal balance of performance and cost-effectiveness for CO₂ capture. For a more detailed description of the case study and these KPIs, please refer to the work of Charalambous et al. [3].

Figure 6 a) illustrates the effectiveness and cost efficiency of different M-TBAPy MOFs in recovering CO₂. Al-TBAPy demonstrates a moderate recovery of 88% with a relatively low nCAC of 175 under dry conditions, indicating a cost-effective solution with a decent recovery level. Ga-TBAPy shows a similar recovery performance to Al-TBAPy but with a slightly higher nCAC. In-TBAPy has a lower recovery rate and a higher nCAC, making it less effective and more expensive. Sc-TBAPy achieves the lowest recovery of 80% and the highest nCAC of 580 under dry conditions. It is important to note that for this specific case study, the M-TBAPy MOFs remain less competitive than the monoethanolamine (MEA) benchmark, which has a nCAC of 100 [3]. Figure 6 b) evaluates the purity of the captured CO₂ relative to the associated cost. The trend reflects that of recovery, with Al-TBAPy achieving the highest purity at 77% and Sc-TBAPy the lowest at 63% under dry conditions. This figure indicates that none of the evaluated materials meet the purity requirement for geological storage, which is set at 96% [3]. This highlights a significant limitation in the current performance of the M-TBAPy MOFs, emphasizing the need for further optimization to reach the required purity levels for effective carbon sequestration. Figure 6 c) examines the productivity of M-TBAPy MOFs against the specific thermal energy required. Similar to previous trends, Al-TBAPy stands out with the highest productivity and the lowest specific thermal energy consumption, indicating efficient CO₂ capture with reasonable energy usage. Productivity declines and specific thermal energy consumption increases in the following order of metals: Ga, In, and Sc. Figure 6 d) addresses the environmental impacts of M-TBAPy MOFs, focusing on the balance between greenhouse gas emissions and the use of material resources (MR:MM). Ideal values are lower on both axes, indicating a smaller environmental footprint. Al and Ga MOFs exhibit the lowest climate change impact and MR:MM, highlighting their environmental efficiency. In-based MOFs show a moderate climate change impact, but are resource-intensive.

Figure 6 clearly demonstrates the impact that water has on the performance of the different MOFs studied here when introduced in the feed. In fact, the presence of water leads to an average decrease in purity of approximately 6% across all MOF structures. This trend aligns with the observed decrease in working capacity under wet conditions, which similarly drops by 6% compared to dry conditions. Interestingly, the recovery rates remain unaffected by the presence of water. The altered composition of the product stream, characterized

by reduced CO₂ content and increased H₂O, results in more intensive energy requirements for regenerating the MOs, thereby causing an increase in the nCAC values. The extent to which different metals are influenced by water varies, largely due to the specific interactions between H₂O, CO₂, and the metal node. For instance, Sc-TBAPy is particularly sensitive, with its nCAC value nearly doubling in the presence of water, whereas Al-TBAPy exhibits a more moderate increase of 25% in its nCAC.

And in the *Discussion* section:

When assessing the cost-effectiveness and scalability of these sorbents, we see that Al-TBAPy consistently performs well across multiple metrics, making it the most balanced and favorable MOF for CO₂ capture in this study. Ga-TBAPy is a close second, offering similar benefits with slightly higher costs. In-TBAPy and Sc-TBAPy are less desirable due to their lower recovery rates and higher costs and environmental impact. Although these physisorbent materials do not match the performance of currently available MEA technologies for practical CO₂ applications, in this study, we highlight their potential and interesting insights, which can be useful for the development of new sorbent materials for CO₂ capture.

Comment 2.4 — The topic taken up by the authors (metal effects on CO₂ uptake in MOFs) is not novel since the first report of such influence dates back to 2008 (10.1021/ja8036096).

Reply: The reviewer is correct that our work is not the first to report on the influence of metals on the CO₂ uptake in general. However, to the best of our knowledge, this is the first experimental study that addresses the specific effect of changes in the metal on the adsorption properties of MOFs formed by stacked pyrene rings (please, see Comments 2.0 - 2.3).

We tried to make this clearer in the revised version:

The impact of metal substitution on the CO₂ capture of different families of MOFs has been previously investigated. [4, 5, 6] In this study, we analyze a particular class, pyrene-based MOFs, in dry and wet conditions, and focus on the influence of these substitutions on the CO₂ uptakes and adsorption properties of these particular MOFs, where the distance between pyrene stacks is tuned depending on the metal incorporated in the structure.

Comment 2.5 — The currently submitted incremental and unoriginal study does not meet the standards of *Nature Communications* and shall not be published.

Besides the uncorrectable lack of novelty, the study also presents deplorably sub-standard technical quality:

Reply: In our reply to Points 2.1 and 2.2, we argue that the reviewer’s opinion on the novelty is factually incorrect.

We thank the reviewer for pointing out some errors in the manuscript. We agree that we should have spotted them, but we would like to emphasize that these are errors in the presentation. Again, we should have been more careful, and we sincerely apologize for making those errors, but these errors in presentation by no means imply a deplorable substandard technical quality of our work.

Comment 2.6 — The aromaticity scheme shown in Figure 1 and Figure S1 is incorrect.

Reply: This has been corrected in the revised version of the manuscript and SI.

Comment 2.7 — The scheme shown in Figure S1 does not match the description of the Synthesis of 1 (both on page S3). The synthesis of 1 and 1' is described in part using a non-standard imperative instead of past tense.

Reply: The reaction scheme has been corrected. We have also rewritten the synthesis procedure of the ligand in the past tense, as shown below and in the revised version of the SI:

Synthesis of 1'. Procedure adapted from Wang et al. [7]. Dioxane (270 mL) was added into a 500 mL three-necked round-bottom flask equipped with a magnetic stirrer and water condenser. Dioxane was put under constant stirring and degassed with N₂ for approximately 1.5 hours. With the N₂ still purging and the dioxane under stirring, 1,3,6,8-tetrabromopyrene (5 g), 4-(ethoxycarbonylphenyl)boronic acid (8.25 g), potassium phosphate tribasic (16.5 g) and tetrakis(triphenylphosphine)-palladium(0) (0.75 g) were inserted into the flask (the solution turned brown). The opening of the round-bottom flask was plugged with a glass plug, and the system was purged with N₂ for an additional 5 minutes. With the help of a heating mantle, the suspension was heated to 90 °C for 48 h to 72 h. The color of the suspended solid should become more yellow as the reaction proceeds. The solution turned black at the end of the reaction. Once the reaction was complete, water (200 mL) was added to the reaction mixture, and the mixture was let to cool down. The reaction mixture was then filtered with a glass Büchner funnel (200 mL), with a medium frit. The yellow solid was collected on the frit and washed with water (2x 100 mL) and acetone (200 mL). Boiling chloroform (300 mL) was then poured onto the glass frit to dissolve the desired product. Methanol (300 mL) was added to the solution which was at room temperature. A light yellow precipitate was formed as methanol was added to the solution. The suspension was let to sit for 30 minutes and the yellow solid was collected with a glass Büchner funnel (200 mL) with a medium filter. The product was dried overnight in a vacuum oven at 70 °C. Approximately 4 g of 1 (i.e., 1,3,6,8-tetrakis(4-(methoxycarbonyl)phenyl)pyrene) should be obtained. The product was analyzed by ¹H-NMR spectroscopy by dissolving it in Chloroform-d.

Synthesis of 1. Procedure adapted from Stylianou et al. [8]. Product 1 (1 g) was dispersed in a solvent mixture THF/dioxane/H₂O (ratio 5/2/2) (100 mL). Concentrated NaOH (20 mL) was added to it. The mixture was then stirred under reflux at 85 °C overnight. Once the reaction was complete, water was added to the suspension, and a clear yellow solution was formed. The solution was stirred at room temperature for 1.5 hours. Using concentrated HCl (32 wt%), the pH of the solution was adjusted to 2. A yellow precipitate was formed, collected by filtration, and washed with water, HCl (1 M), and diethyl ether. The solid was then dried under vacuum in a ventilated oven at 70 °C overnight. Once the product was properly dry, boiling DMF was added to the yellow powder. The solution was left under stirring and filtered before cooling to room temperature. Once the solution had cooled down, dichloromethane (300 mL) was added, and a yellow solid was formed. The mixture was filtered with a glass Büchner funnel (200 mL) with a medium filter, and the powder was dried in a vacuum oven at 70 °C overnight to obtain 1' (TBAPy) (0.78 g). The product was analyzed by ¹H-NMR spectroscopy by dissolving it in DMSO-*d*₆.

Comment 2.8 — Yields of MOF materials syntheses are not reported.

Reply: We have now added all yields in the synthesis section of the revised version of the manuscript.

Comment 2.9 — The statement claiming that Ga-TBAPy has not been reported is incorrect: 10.1039/D3TA05297D.

Reply: We thank the reviewer for bringing this interesting work to our attention — we have also cited it in the revised version of the manuscript.

The first author synthesized Ga-TBAPy as part of her MSc thesis in 2021 [9]. In the revised version, we refer to both reports to emphasize that the synthesis of this MOF has been done independently. We also removed any reference to novelty, and cite both references:

In this work, we synthesize four different pyrene-based MOFs, with general formula $M_2(OH)_2(TBAPy)$ (with $M = Al(III), Ga(III), Sc(III), \text{ and } In(III)$). The synthesis of the ligand is based on two reported procedures [7, 8] (Supplementary Note 1). The syntheses of Al-TBAPy [2], Sc-TBAPy [10], In-TBAPy [8], and Ga-TBAPy [9, 11] are also reported.

Comment 2.10 — The discovery of the monoclinic phase of Ga-TBAPy from electron diffraction is interesting but not given much attention. The space group is inconclusively determined as $P2(1)/n$ or $P2/c$ in Supplementary Note 5.

Reply: We thank the reviewer for noticing the typo in the space group. The data shows it is $P2/c$, and this has been corrected in the revised version of the manuscript and SI:

$C_{22}H_{12}GaO_5$ ($M = 426.056$ g/mol): monoclinic, space group $P2/c$ (no. 13), $a = 11.1(4)$ Å, $b = 15.1(3)$ Å, $c = 12.3(4)$ Å, $\beta = 90.12(7)^\circ$, $V = 2052(106)$ Å³, $Z = 4$, $T = 298$ K, $\mu(\text{electrons}) = 0.000$ mm⁻¹, $D_{\text{calc}} = 1.379$ g/cm³, 26518 reflections measured ($0.2^\circ \leq 2\Theta \leq 1.8^\circ$), 3614 unique ($R_{\text{int}} = 0.2644$, $R_{\text{sigma}} = 0.1629$) which were used in all calculations. The final R_1 was 0.2425 ($I > 2\sigma(I)$) and wR_2 was 0.5356 (all data). CCDC 2327953.

Comment 2.11 — Rietveld refinement of the powder XRD pattern of Ga-TBAPy accounting for both the orthorhombic and the monoclinic phases (Figure S7) is not satisfactory as there is still a clear difference between the model and the data, which the authors explicitly admit (“data quality does not allow us to refine the actual structure”). The comparison between the calculated and experimental patterns (Figure S8 and Figure 3d) is equally inconclusive. Therefore, assertions about Ga-TBAPy (“structural distortion”, “structural rearrangement”, “lattice distortion”, “Ga-TBAPy [...] no longer shows the perfect stacking of the TBAPy ligand”, “rotations of the benzoate groups due to stress during synthesis and activation”) may be inaccurate. Presence of phases other than o-Ga-TBAPy and m-Ga-TBAPy should equally not be excluded.

Reply: We understand the reviewer’s concern, and we agree with the statement. The data quality and complexity of the Ga-TBAPy structure do not allow us to fully refine the structure, and the presence of additional phases might be an option. We have made this clearer in the revised version of the manuscript:

Confirming our hypothesis of two phases present in the activated Ga-TBAPy, Rietveld refinement of the synchrotron *ex-situ* PXRD measurements reveals that, although an orthorhombic $Cmmm$ -phase is present a monoclinic mcl -phase co-exists (Supplementary Note 3). However, given the complexity and quality of the data, additional phases may not be excluded.

Moreover, the assertions to which the reviewer refers to are not made purely based on the Rietveld refinements, but we take into account data from isotherms, variable temperature PXRD, and reports from literature [8, 12] to support these hypotheses.

Comment 2.12 — The hkl tags are incorrect in all Rietveld refinements (Figures S4-S7).

Reply: The *hkl* tags have been corrected accordingly, as shown in Figure S3:

Comment 2.13 — The interpretation of the 77 K N₂ isotherm of Al-TBAPy (Figure S10a) as type II is incorrect.

Reply: We have now corrected this in the revised version of the main text:

The pore volumes of the materials are evaluated by carrying out N₂ adsorption isotherms at 77 K (Supplementary Note 7), and all materials demonstrate permanent microporosity, as shown by the type I N₂ isotherms [13] (Figure S6 a).

Comment 2.14 — The claim that Al(0.50)Sc(0.50)-TBAPy features a scheme of alternating Al and Sc atoms in its structure (Figure S3a) is dubious. Any other arrangement of these atoms may equally be possible.

Reply: We understand the reviewer’s concern. We have tested four different Al_{0.50}Sc_{0.50}-TBAPy structures, each with a unique arrangement of the metal. The revised version of the manuscript now reads:

To assess the impact of local structural modifications and nonparallel pyrene stacks caused by different Al and Sc arrangements, we generate *in silico* CIFs with different Al and Sc configurations (Supplementary Note 19). The obtained PXRD patterns of the simulated structures are compared to the experimental PXRDs of Al_{0.50}Sc_{0.50}-TBAPy (Figure S18 a). The different computational results do not show significant differences in diffraction data, and match relatively well the experimental Al_{0.50}Sc_{0.50}-TBAPy pattern. The CO₂ and N₂ uptakes at 40 °C, as well as N₂ isotherms at 77 K and pore volume measurements for the Al_{0.50}Sc_{0.50}-TBAPy MOF are also conducted (Figure S6 and S18). The results follow the expected trend, and we see that the CO₂ uptake of the mixed-metal MOF lies between the pure Al- and Sc-TBAPy ones. The computed isotherms of Al_{0.50}Sc_{0.50}-TBAPy with different Al *vs* Sc arrangements slightly over-predict the experimentally measured isotherm (Figures S18 b and c), but show that overall, varying the arrangement of the Al and Sc metals in the structure, does not have a significant impact on the uptake of the materials. Finally, the experimental H₂O vapor adsorption isotherm at 40 °C shows that this structure starts adsorbing H₂O between approximately 50-60% relative humidity, which falls between the values of the pure Al- and Sc-TBAPy structures (Figure S18 d).

Comment 2.15 — In Figure 4, CO₂ adsorption isotherms were plotted on an unconventional log-log scale. Moreover, the x-axes should be labeled as “Pressure” and not “log(Pressure)” since log(0.01) = -2, log(0.1) = -1, and log(1) = 0. The y-axes are equally mislabeled.

Reply: Corrected.

Comment 2.16 — In the discussion of Figure 4, the experimental CO₂ uptake of Ga-TBAPy is thought to lie in between of the trends computed for o-Ga-TBAPy and m-Ga-TBAPy. However, experimental CO₂ uptake of Al-TBAPy is not accurately computed neither, despite the MOF occurring in its orthorhombic phase only.

Reply: We have re-run all the CO₂ and N₂ isotherms with an improved model as explained in the updated *Methods* section, highlighted below. We have updated the revised version of the manuscript accordingly, where we show a much closer match for all the materials. The revised version of the manuscript now reads:

In the Henry regime, the uptakes of the simulated orthorhombic structures follow the trend expected if one looks solely at the inter-aromatic spacing: Al-TBAPy > Ga-TBAPy > In-TBAPy > Sc-TBAPy. Experimentally, we see a similar ranking: Al-TBAPy \approx Ga-TBAPy > In-TBAPy \approx Sc-TBAPy. Interestingly, in the low-pressure regime, the experimental CO₂ isotherms of Ga- and In-TBAPy align more closely with the computational data calculated from the monoclinic CIFs (Supplementary Note 11), while at high pressures, they correspond more accurately to the orthorhombic data (Figure 4 a). In fact, the simulated monoclinic structures reach the maximum CO₂ loading at much lower pressures compared to their orthorhombic counterparts, which may be ascribed to their lower pore volume. This may suggest that upon CO₂ loading, the structures may distort from monoclinic to orthorhombic. The phase transitions upon guest loading have been carefully investigated in the original publication of In-TBAPy, for which re-immersing the activated MOF (i.e., monoclinic structure) in DMF or dioxane, leads to a reversible transformation back to its orthorhombic structure [8]. We believe a similar structural flexibility occurs as the In- and Ga-TBAPy MOFs are loaded with CO₂ (Supplementary Note 12).

And in the revised version of the *Methods* section and SI, we now have:

Computational Methods. In this study, the DFT calculations were conducted using the Quickstep code of the CP2K package (version 9.1) [14]. This code is an efficient DFT implementation for large and complex structures by exploiting the mixed Gaussian and plane waves (GPW) method alongside pseudopotentials, optimizing the wave function with the orbital transformation (OT) technique. We employed double-zeta DZVP-MOLOPT-SR contracted basis sets and GTH pseudopotentials to represent the electronic wave function. The multigrid used for plane waves had a 4-level structure with a primary cutoff of 600 Ry, a relative cutoff of 50 Ry, and a progression factor of 3. For the exchange-correlation energy, we applied the Perdew-Burke-Ernzerhof (PBE) [15] functional with the DFT-D3(BJ) [16] model to account for many-body dispersion interactions. The selection of integration grid and DFT functional was based on the research of Ongari et al. [17], who provided optimized DFT settings for accurate and efficient high-throughput computational analysis of covalent organic frameworks (COFs) and MOFs.

And

The RASPA molecular simulation software for adsorption and diffusion in nanoporous materials[18] was used to perform MC simulations. The optimized framework geometries were kept rigid in all classical simulations. We considered van der Waals and electrostatic interactions to describe the energy surface, represented respectively by the Lennard-Jones (LJ) potential and Coulomb interactions. Periodic boundary conditions were employed with a cutoff radius of 12.8 Å, including tail corrections to remedy the truncation. density-derived electrostatic and chemical (DDEC) method is used to compute the partial charges on the atoms of the MOF frameworks.[19] The Ewald summation technique was used to model Coulomb interaction[20]. The TraPPE FF[21] was selected to model gas-gas interactions for CO₂ and N₂. The dispersion interactions of the framework and the gases were modeled with Lennard-Jones potentials. The Lennard-Jones parameters are refined from the Universal Force Field (UFF)[22] according to the experimental CO₂ adsorption isotherm[23].

Unified workflows were used for several parts of this study to ensure the reproducibility and direct comparability of computed data. The Automated Interactive Infrastructure and Database for Computational Science (AiiDA)[24] was

employed to orchestrate the different steps, managing the interaction of different codes and providing automation and similarity of the calculations. The workflows are published and maintained as the "AiiDA-LSMO" plugin on GitHub[25].

Single-Component Isotherm Workflow. The "Isotherm" work chain in the "AiiDA-LSMO" plugin was used to generate pure CO₂, and N₂ adsorption isotherms. The adsorption isotherms were simulated in the Grand Canonical ensemble (GCMC ensemble).[26] Here, 15000 cycles were used for equilibration and 15000 cycles for production. Simulations at subsequent pressure points were performed starting from the restart file of the previous pressure step, thus reducing the number of cycles necessary for initialization.

Binding Site Workflow. The minimum energy configuration of adsorbate molecules (CO₂, N₂, H₂O) in the different frameworks was determined using the Monte Carlo (MC) simulations. The force field (FF) parameters were kept the same as mentioned above, and 10⁴ MC cycles were used to determine the molecule position. Combined with density maps, we recorded the energies from FF simulations and subsequently optimized those with adsorbate molecules inserted configurations with minimal energies in DFT simulation. Both MOF structures and adsorbate molecules are fully relaxed. Then we performed basis set superposition error (BSSE) correction simulation [27] to cancel the energy difference resulting from basis functions overlap. The adsorption binding energy can be calculated as:

$$E_{BD} = E_{\text{system}} - E_{\text{MOF}} - E_{\text{adsorbate}},$$

where E_{system} is the BSSE corrected energy of the relaxed MOF and adsorbates, and E_{MOF} is the energy of the relaxed MOF structure without an adsorbate.

Orthorhombic M-TBAPy (with M = Al, Ga, In, and Sc). The computational models of M-TBAPy were derived using the reported coordination file for orthorhombic Al-TBAPy [2], swapping Al for Ga, In, and Sc, respectively. The cell parameters were fully relaxed with the cell optimization simulation in DFT, except for reported Al-TBAPy.

Monoclinic M-TBAPy (with M = Ga, and In). As no experimental information for the orthorhombic cell parameters was available, the framework was created based on the crystallographic information provided by Stylianou et al. [8]. The monoclinic model was built based on crystallographic information of the micro-electron diffraction (microED/3D-ED) Ga-TBAPy structure (see next paragraph). Then, we replaced Ga with In, and the cell parameters were fully relaxed with the cell optimization simulation in DFT.

Orthorhombic Al_{0.50}Sc_{0.50}-TBAPy. The computational model of Al_{0.50}Sc_{0.50}-TBAPy was constructed based on the crystallographic data of orthorhombic Al-TBAPy. A mixed-metal framework was created by replacing half of the Al atoms with Sc atoms. The cell parameters were fully relaxed with the cell optimization simulation in DFT.

Comment 2.17 — The authors discuss the metal effects on CO₂ uptake in pyrene-based MOFs without considering the density of the studied materials. It is striking that the MOF constructed from the lightest metal element (Al) displayed the highest uptake [expressed in mmol(CO₂)/g(MOF)], while the MOF constructed from the heaviest metal element (In) displayed the lowest uptake. An explanation that different observed uptakes are just the consequence of different MOF densities and not different adsorbent–adsorbate interactions was not considered.

Reply: This is a good point. Given the different unit cell parameters of the MOFs (i.e., orthorhombic and monoclinic) (Table 1), which largely affect the density and cell volume of

the structures (Tables S1), we believe that density-based normalization is not appropriate in this case.

Table 1: Computational cell parameters obtained for the different monoclinic and orthorhombic structures.

Computational Cell Parameters	Orthorhombic Al-TBAPy	Orthorhombic Sc-TBAPy	Monoclinic Ga-TBAPy	Orthorhombic Ga-TBAPy	Monoclinic In-TBAPy	Orthorhombic In-TBAPy
a (Å)	30.78	30.96	11.10	30.51	10.90	30.75
b (Å)	6.632	7.230	15.10	6.649	15.97	7.120
c (Å)	15.59	16.00	12.30	15.97	12.39	15.93

We therefore normalized the data using the molecular weight of the structures:

Here, we present the uptakes in $\text{mmol}_{[Adsorbate]}/\text{mmol}_{[MOF]}$ to emphasize that our results are not due to the weight of the metal incorporated in the structure. The factors used for conversion from mmol/g to mmol/mmol can be found in Supplementary Note 10.

Comment 2.18 — Experimental values of isosteric heats of CO_2 adsorption, now standard in the literature on carbon capture, are not determined.

Reply: We have collected additional CO_2 isotherms at 40 °C and 55 °C to calculate the isosteric heat of adsorption. We have also updated the revised version of the SI accordingly:

Experimental CO_2 adsorption isotherms were collected at 25, 40, and 55 °C and fitted to a dual-site Langmuir model following:

$$q = q_{sat,1} \frac{b_1 P}{1 + b_1 P} + q_{sat,2} \frac{b_2 P}{1 + b_2 P} \quad (1)$$

where q corresponds to the adsorbed amount in mmol/g , $q_{sat,1}$ is the adsorption capacity for site 1, b_1 is the Langmuir parameter for site 1 ($q_{sat,2}$ and b_2 are equivalent for site 2) and P is the pressure in Pa.

Subsequently, the Clausius-Clapeyron equation (2) was used to calculate the isosteric enthalpy of adsorption (i.e., Q_{st}) for CO_2 :

$$\ln(P) = -\frac{Q_{st}}{R} \left(\frac{1}{T} \right) + c \quad (2)$$

where R is the ideal gas law constant, T is the temperature, and c is a constant.

We have updated the revised version of the manuscript as follows:

To further assess the performance of these materials for carbon capture applications, we also report the experimental and computational isosteric heats of CO_2 adsorption (Q_{st}) (Supplementary Note 15). The experimental Q_{st} is calculated using variable temperature adsorption isotherms at 25, 40, and 55 °C, and the values at zero loading follow a similar trend as the computational Q_{st} . These results are further supported by the DFT-calculated binding energies for the minimum energy configurations of the orthorhombic Al- and Sc-TBAPy and monoclinic structures of Ga- and In-TBAPy.

Comment 2.19 — In the discussion of TGA as well as that of atomic (or ionic, the text confuses the two) radii, Sc-TBAPy is somehow omitted as if it had not made part of the series.

Reply: We have corrected this in the revised version:

Many factors are impacted if we change the metal. The most prominent one is the effective ionic radius: $r_{Al^{3+}} < r_{Ga^{3+}} < r_{Sc^{3+}} < r_{In^{3+}}$ [28]. Such an increase in ionic radii lowers the spatial overlap between the oxygen orbitals of the carboxylate groups of the ligand and the metal, resulting in weaker bonds, larger metal-oxygen ($M-O_{Ligand}$) distances, and thus higher inter-aromatic spacing between ligands. However, although Sc(III) has a smaller ionic radius than In(III), its inter-aromatic spacing is bigger. By looking at the M-O-M angle, we notice that In-TBAPy has a much smaller angle compared to the other MOFs, which consequently leads to the smaller inter-aromatic spacing observed in this structure.

Comment 2.20 — The inclusion of $Al_{0.50}Sc_{0.50}$ -TBAPy at the end of the manuscript also seems as an afterthought.

Reply: We have added a paragraph featuring the interest of studying this structure in the *Introduction* and designated a separate chapter in the *Results* section of the revised version. The new version of the *Introduction* now reads:

Moreover, we experimentally and computationally investigate the performance of a mixed-cation TBAPy-based MOF. We explore different Al *vs* Sc metal arrangements within the same structure, which induce nonparallel pyrene stacks, and thus affect the critical adsorption sites of CO_2 . Here, we aim to elucidate how these local structural modifications influence the uptake of CO_2 , N_2 , and H_2O , and thus provide interesting insights into the adsorption behavior of these materials.

And we have also added an additional paragraph in the *Fine-Tuning Uptakes: A Mixed-Cation-TBAPy MOF* section, as detailed in Comment 2.14.

Reviewer 3

This manuscript concerns design of better CO₂ capture physisorbents. In prior work, a design paradigm had been proposed that spaces pyrene linkers in MOFs at roughly 0.7 nm to create pockets favorable for CO₂. This work studies such systems with four metal variations that allow for effects of small changes in spacing to be studied with regards to CO₂ separations. The authors report mixed phases in some examples showing a more complex scenario exists than predicted theoretically. Experimental CO₂ binding is studied and the binding sites extracted computationally. I feel the work is generally well-executed but there is a main point I feel that needs to be addressed. The entire point of the pyrene paradigm is that the binding sites in such systems would allow physisorptive CO₂ binding in wet gases. This point is mentioned in the introduction but not addressed in the experiments. There are numerous studies on subtle structural variation of structure by metal exchange in MOFs including for CO₂ capture. Given the paradigm has already been presented for this exact family, I do not feel that metal variation and its effects is itself sufficient novelty to merit such a high impact journal.

Comment 3.1 — If the authors can address the water co-sorption, this can be reconsidered. It is still suitable for a high impact chemistry-specific journal.

Reply: We agree with the reviewer's comment and appreciate the helpful feedback. Based on the suggestions, we have conducted additional experiments and updated the manuscript accordingly to more explicitly emphasize the novel aspects of our study, including adding the experimental investigation of metal variation effects on water vapor adsorption isotherms. Moreover, we also tested our materials in simulated dry and wet conditions, as detailed in Comment 2.3.

References

- [1] Archit Datar, Matthew Witman, and Li-Chiang Lin. Improving computational assessment of porous materials for water adsorption applications via flat histogram methods. *The Journal of Physical Chemistry C*, 125(7):4253–4266, 2021.
- [2] Peter G Boyd, Arunraj Chidambaram, Enrique García-Díez, Christopher P Ireland, Thomas D Daff, Richard Bounds, Andrzej Gładysiak, Pascal Schouwink, Seyed Mo-hamad Moosavi, M Mercedes Maroto-Valer, et al. Data-driven design of metal–organic frameworks for wet flue gas co2 capture. *Nature*, 576(7786):253–256, 2019.
- [3] Charithea Charalambous, Elias Moubarak, Johannes Schilling, Eva Sanchez Fernandez, Jin-Yu Wang, Laura Herraiz, Fergus Mcilwaine, Shing Bo Peh, Matthew Garvin, Kevin Maik Jablonka, Seyed Mo-hamad Moosavi, Joren Van Herck, Aysu Yurdusen Ozturk, Alireza Pourghaderi, Ah-Young Song, Georges Mouchaham, Christian Serre, Jeffrey A. Reimer, André Bardow, Berend Smit, and Susana Garcia. A holistic platform for accelerating sorbent-based carbon capture. *Nature*, 632(8023):89–94, July 2024. ISSN 1476-4687. doi: 10.1038/s41586-024-07683-8. URL <http://dx.doi.org/10.1038/s41586-024-07683-8>.
- [4] Stephen R Caskey, Antek G Wong-Foy, and Adam J Matzger. Dramatic tuning of carbon dioxide uptake via metal substitution in a coordination polymer with cylindrical pores. *Journal of the American Chemical Society*, 130(33):10870–10871, 2008.
- [5] Hui Zhang, Chunli Shang, Li-Ming Yang, and Eric Ganz. Elucidation of the underlying mechanism of co2 capture by ethylenediamine-functionalized m2 (dobpdc)(m= mg, sc-zn). *Inorganic chemistry*, 59(22):16665–16671, 2020.
- [6] Ankit Agrawal, Mayank Agrawal, Donguk Suh, Shubo Fei, Amer Alizadeh, Yunsheng Ma, Ryotaro Matsuda, Wei-Lun Hsu, and Hirofumi Daiguji. Augmenting the carbon dioxide uptake and selectivity of metal–organic frameworks by metal substitution: Molecular simulations of lmof-202. *ACS omega*, 5(28):17193–17198, 2020.
- [7] Timothy C Wang, Nicolaas A Vermeulen, In Soo Kim, Alex BF Martinson, J Fraser Stoddart, Joseph T Hupp, and Omar K Farha. Scalable synthesis and post-modification of a mesoporous metal-organic framework called nu-1000. *Nature protocols*, 11(1):149–162, 2016.
- [8] Kyriakos C Stylianou, Romain Heck, Samantha Y Chong, John Bacsá, James TA Jones, Yaroslav Z Khimyak, Darren Bradshaw, and Matthew J Rosseinsky. A guest-responsive fluorescent 3d microporous metal-organic framework derived from a long-lifetime pyrene core. *Journal of the American Chemical Society*, 132(12):4119–4130, 2010.
- [9] Thomas F. Willems, Chris H. Rycroft, Michael Kazi, Juan C. Meza, and Maciej Haranczyk. Understanding & application driven design of metal-organic frameworks for carbon capture. *Infoscience EPFL Scientific Publications*, 2021. URL <https://infoscience.epfl.ch/record/311826>.
- [10] F Pelin Kinik, Andres Ortega-Guerrero, Fatmah Mish Ebrahim, Christopher P Ireland, Ozge Kadioglu, Amber Mace, Mehrdad Asgari, and Berend Smit. Toward optimal photocatalytic hydrogen generation from water using pyrene-based metal-organic frameworks. *ACS applied materials & interfaces*, 13(48):57118–57131, 2021.
- [11] Michelle Åhlén, Yi Zhou, Daniel Hedbom, Hae Sung Cho, Maria Strømme, Osamu Terasaki, and Ocean Cheung. Efficient sf 6 capture and separation in robust gallium-and vanadium-based metal–organic frameworks. *Journal of Materials Chemistry A*, 11(48):26435–26441, 2023.

- [12] Kyriakos C Stylianou, Jeremy Rabone, Samantha Y Chong, Romain Heck, Jayne Armstrong, Paul V Wiper, Kim E Jelfs, Sergey Zlatogorsky, John Bacsá, Alec G McLennan, et al. Dimensionality transformation through paddlewheel reconfiguration in a flexible and porous zn-based metal–organic framework. *Journal of the American Chemical Society*, 134(50):20466–20478, 2012.
- [13] Marc D Donohue and Grigoriy L Aranovich. Classification of gibbs adsorption isotherms. *Advances in colloid and interface science*, 76:137–152, 1998.
- [14] Thomas D. Kühne¹, Marcella Iannuzzi, Mauro Del Ben, Vladimir V. Rybkin, Patrick Seewald, Frederick Stein, Teodoro Laino, Rustam Z. Khaliullin, Ole Schütt, Florian Schiffmann, Dorothea Golze, Jan Wilhelm, Sergey Chulkov, Mohammad Hossein Bani-Hashemian, Valéry Weber, Urban Borštnik, Mathieu Taillefumier, Alice Jakobovits, Alfio Lazzaro, Hans Pabst, Tiziano Müller, Robert Schade, Manuel Guidon, Samuel Andermatt, Nico Holmberg, Gregory K. Schenter, Anna Hehn, Augustin Bussy, Fabian Belleflamme, Gloria Tabacchi, Andreas Glöß, Michael Lass, Iain Bethune, Christopher J. Mundy, Christian Plessl, Matt Watkins, Joost VandeVondele, Matthias Krack, and Hütter) Hutter. Cp2k: An electronic structure and molecular dynamics software package - quickstep: Efficient and accurate electronic structure calculations. *The Journal of Chemical Physics*, 152(19):194103–1–194103–47, 2020.
- [15] John P. Perdew, Kieron Burke, and Matthias Ernzerhof. Generalized gradient approximation made simple. *Phys. Rev. Lett.*, 77:3865–3868, 1996.
- [16] Stefan Grimme, Stephan Ehrlich, and Lars Goerigk. Effect of the damping function in dispersion corrected density functional theory. *Journal of Computational Chemistry*, 32(7):1456–1465, 2011.
- [17] Daniele Ongari, Aliaksandr V Yakutovich, Leopold Talirz, and Berend Smit. Building a consistent and reproducible database for adsorption evaluation in covalent-organic frameworks. *ACS central science*, 5(10):1663–1675, 2019.
- [18] David Dubbeldam, Sofia Calero, Donald E. Ellis, and Randall Q. Snurr. Raspa: molecular simulation software for adsorption and diffusion in flexible nanoporous materials. *Molecular Simulation*, 42(2):81–101, 2016.
- [19] Thomas A Manz and David S Sholl. Chemically meaningful atomic charges that reproduce the electrostatic potential in periodic and nonperiodic materials. *Journal of Chemical Theory and Computation*, 6(8):2455–2468, 2010.
- [20] P. P. Ewald. Die berechnung optischer und elektrostatischer gitterpotentiale. *Annalen der Physik*, 369(3):253–287, 1921.
- [21] Katie A Maerzke, Nathan E Schultz, Richard B Ross, and J Ilja Siepmann. Trappe-ua force field for acrylates and monte carlo simulations for their mixtures with alkanes and alcohols. *The Journal of Physical Chemistry B*, 113(18):6415–6425, 2009.
- [22] A. K. Rappe, C. J. Casewit, K. S. Colwell, W. A. Goddard, and W. M. Skiff. Uff, a full periodic table force field for molecular mechanics and molecular dynamics simulations. *Journal of the American Chemical Society*, 114(25):10024–10035, 1992.
- [23] Yutao Li, Xin Jin, Elias Moubarak, and Berend Smit. A refined set of universal force field parameters for some metal nodes in metal-organic frameworks. *ChemRxiv*, 2024. doi: 10.26434/chemrxiv-2024-btm9c. Preprint.
- [24] Giovanni Pizzi, Andrea Cepellotti, Riccardo Sabatini, Nicola Marzari, and Boris Kozinsky. Aiida: automated interactive infrastructure and database for computational science. *Computational Materials Science*, 111:218–230, 2016. ISSN 0927-0256. doi: <https://doi.org/10.1016/j.commatsci.2015.09.013>. URL <https://www.sciencedirect.com/science/article/pii/S0927025615005820>.

- [25] T. LSMO and AiiDA team. An aiiDA workflows for the lsmo laboratory at epfl, 2023. URL <https://github.com/lsmo-epfl/aiida-lsmo>. Accessed: 2023.
- [26] D. Frenkel and B. Smit. *Understanding Molecular Simulations: from Algorithms to Applications*. Academic Press, San Diego, 3rd edition, 2023.
- [27] S. F. Boys and F. Bernardi. Calculation of small molecular interactions by differences of separate total energies – some procedures with reduced errors. *Molecular Physics*, 19:553—566, 2021.
- [28] Thomas Devic and Christian Serre. High valence 3p and transition metal based mofs. *Chemical Society Reviews*, 43(16):6097–6115, 2014.

Response to the Reviewers

Reviewer 1

I'm OK with it being published in the current form.

Reply: We thank the reviewer for the comment and helpful feedback during the revision process.

Reviewer 2

The authors have successfully addressed all the comments raised during the review. The intensity and fiery nature of the review process underscore the importance of this work. We recommend the manuscript for publication.

Reply: We thank the reviewer for the positive comment and helpful feedback.

Reviewer 3

My prior comment related to the critical absence of CO₂/water co-sorption data. The authors have added water sorption. They have not experimentally addressed the point of water/CO₂ co-adsorption. This is not an exceptional request for a work making the claims in this work. The use of the Prisma platform to extract process level metrics for “wet” systems is not a substitute for water/CO₂ breakthrough experiments. In fact, to extrapolate process metrics when a crucial experiment has not been done sets a poor precedent. Given that the family of compounds has been reported in Nature previously, I do not feel this work is a significant improvement in terms of carbon capture, certainly not at the level to merit such a high profile venue. If this was for JACS or Angewandte Chemie, I would still expect to see the dynamic column breakthrough in wet gas as the claim is central to the novelty and impact of this work.

Reply: We understand the value breakthrough experiments can play when assessing materials for industrial applications, particularly in wet conditions. In fact, in our original study that motivated this work (Nature 576 (7786), 253 (2019) <http://dx.doi.org/10.1038/s41586-019-1798-7>), we used breakthrough experiments to demonstrate that one of the materials outperformed commercial materials.

However, the focus of this study is elucidating the mechanistic role of the metal in tuning the inter-aromatic spacing of this class of MOFs, their crystallization, and subsequent CO₂ and/or H₂O adsorption behavior. Interestingly, the behavior is much more complex than we originally assumed. However, this complexity negatively impacts the potential performance for carbon capture. One can already see this from the pure component isotherms, and our PrISMa platform calculations further confirm that the carbon capture performance of these materials is not superior. We do not expect that actual breakthrough experiments would change this conclusion. As these breakthrough experiments do not provide any insights into the mechanistic role of the metal, we respectfully disagree with the reviewer that these breakthrough data are essential for this work.